# Continuous 3D printing from one single droplet

Yu Zhang[1,2,5], Zhichao Dong [2,3,5], Chuxin Li[3], Huifeng Du [4], Nicholas X. Fang[4], Lei Wu [1,2✉] & Yanlin Song [1,2✉]

3D printing has become one of the most promising methods to construct delicate 3D structures. However, precision and material utilization efficiency are limited. Here, we propose a one-droplet 3D printing strategy to fabricate controllable 3D structures from a single droplet ascribing to the receding property of the three-phase contact line (TCL) of the resin droplet. The well-controlled dewetting force of liquid resin on the cured structure results in the minimization of liquid residue and the high wet and net material utilization efficiency in forming a droplet into a 3D structure. Additionally, extra curing induced protruding or stepped sidewalls under high printing speed, which require high UV intensity, can be prevented. The critical is the free contact surface property of the droplet system with the introduction of the receding TCL, which increased the inner droplet liquid circulation and reduces the adhesion properties among the liquid resin, cured resin, and resin vat.

[1] Key Laboratory of Green Printing, Institute of Chemistry, Chinese Academy of Sciences, Zhongguancun North First Street 2, 100190 Beijing, P.R. China. [2] University of Chinese Academy of Sciences, Beijing 100049, P. R. China. [3] CAS Key Laboratory of Bio-inspired Materials and Interfacial Sciences, Technical Institute of Physics and Chemistry, Chinese Academy of Sciences, Beijing 100190, P.R. China. [4] Department of Mechanical Engineering, Massachusetts Institute of Technology, Cambridge, MA 02139, USA. [5] The authors contributed equally: Yu Zhang, Zhichao Dong. ✉email: wulei1989@iccas.ac.cn; ylsong@iccas.ac.cn

Additive manufacturing or 3D printing rapidly turns computer-aided designs into complex 3D objects without the use of the molds, dies or lithographic masks that are needed for conventional manufacturing[1,2]. 3D printing based on photocuring, which allows solidification of a 3D model at the curing interface[3–6], has become a promising technique with a variety of applications, such as in biomimetic investigations[7–11], microfluidics[12–14], sensors, and shape-morphing systems[15–19]. Although effective in constructing fine structures, compared with fused deposition modeling (FDM), the existing digital light processing (DLP), stereolithography (SLA) printing and volumetric additive manufacture technologies have low wet material utilization efficiency (the weight ratio of the just printed structure to the initial liquid resin) and net material utilization efficiency (the weight ratio of the acquired dry structure to the initial liquid resin). Uncured resin needs to cover the whole tank with an excessive quantity before the printing process, which not only increases the resin cost but also leads to resin waste. Also, as the UV resin is exothermic, heat dissipation is not sufficient for continuous printing process[20], especially for high-speed printing that requires high UV intensity. Accompanying with the residual resin on the cured structure surface, and the continuous irradiation of the resin under exciting light including the afterglow of the UV projector, extra curing or printing instability occurs which will decrease the 3D printing resolution (Supplementary Figs. 1-3). The adhesion and residue of liquid resin in the vat and on the cured solid surface limit high-resolution construction and increase the cost, especially when using expensive resins.

From the interfacial perspective, the chemical composition and surface roughness of the substrate greatly influence the three-phase contact line (TCL) dynamics[21,22]. Based on observations of natural lotus and pitcher plant surfaces, air or liquid trapped at the surface can greatly reduce the interfacial adhesion on the substrate, resulting in the sphere contact mode of a liquid droplet or the slipping phenomenon for liquids contacting such surfaces[23,24]. Inspired by these phenomena, we demonstrate an interfacial manipulation method to manufacture 3D structures from a single droplet with high wet and net material utilization efficiency. A curing interface with both low liquid resin adhesion and low cured resin adhesion is employed in this system to endow the 3D printing process with a retracting TCL. The amount of residual resin is efficiently reduced during the printing process, and the resin utilization efficiency is significantly improved. Besides, extra curing under high printing speed which requires high UV intensity is prevented. The key is the free contact surface property of the droplet system through the introduction of receding TCL, which increases the inner droplet liquid circulation and reduces the adhesion properties among the liquid resin, cured resin, and resin vat. The devised process, which combines UV curing and droplet TCL dewetting, can be used to effectively grow 3D structures from a single resin droplet with improved 3D printing efficiency and precision.

## Results

**One-droplet 3D printing configuration.** Figure 1 shows a schematic and corresponding time-sequence images of the proposed one-droplet 3D printing process. As shown in Fig. 1a, a single droplet of liquid resin is deposited on the curing interface (Step I), which is the upper surface of the bottom of the resin vat, to realize the continuous printing process. Before the printing process, the supporting plate contacts the curing window (substrate), with the resin droplet in between (Step II). By continuously projecting UV illumination patterns onto the curing interface and elevating the supporting plate at a constant speed, liquid resin can be cured into solid resin that exhibits the UV

pattern at the curing interface (Step III). Solidified 3D structures are thus formed from the liquid resin droplet. Simultaneously, during the printing process, the TCL of the liquid resin droplet recedes with the consumption of the liquid resin through continuous UV curing. Finally, the liquid droplet is cured into a desired 3D solidified structure (Step IV) without residue on the substrate. The resin droplet's continuous receding TCL on the curing interface facilities the printing process in a high resin utilization efficiency. As shown in Fig. 1b–e, a 24-mm-long cured grid structure with a cylindrical shape is printed with a wet resin utilization efficiency of 99.6%. The remaining 0.4% of liquid resin is left on the supporting plate due to the adhesion between liquid resin and supporting plate (as displayed in Fig. 1d).

**Curing interface of one-droplet 3D printing.** A schematic of the one-droplet printing process with bottom-up UV illumination is shown in Fig. 2a. The self-built 3D printing setup involves a UV projector, a UV-transparent curing interface, and an aluminum supporting plate mounting on a programmable moving platform from bottom to up based on our previous investigations[25,26]. The underside of the supporting plate is in contact with the curing surface through penetration of the center of the resin droplet and can be elevated by the programmable moving platform. By mounting the supporting plate on a load cell, the adhesive force generated during the one-droplet printing process can be measured. The light pattern is set as a round shape with a diameter of 1 mm (Fig. 2a inset). The continuous elevation speed of the supporting plate is at most 200 μm/s. To realize one-droplet 3D printing, the essential requirement for the curing interface is that the TCL of the liquid resin can recede on it. The maximum velocity (Fig. 2a) on the curing interface for smooth droplet retraction $v_m$ scales as $v^*\theta_E{}^3/9\sqrt{3}l$[27], where $v^*$ is the threshold capillary velocity and scales as $\mu/\gamma$, $l$[28] typically ranges from 15 to 20, and $\theta_E$ is the equal contact angle. A larger equal contact angle means a larger upper limit of the threshold velocity. Reducing the surface energy through adjusting the chemical compositions, or modifying the surface with micro/nanostructured morphology, not only provides the surface with higher liquid repellency (a larger contact angle) but also enhances the droplet movement property. A fluorinated flat surface is a typical surface with reduced surface energy by grafting fluorinated molecules onto the surface[29], on which the droplet TCL can recede during the evaporation process. Superamphiphobic surfaces are typical surfaces that employ micro/nanoscale structures to effectively entrap an air layer to reduce the surface energy and repel wetting, which facilitates the shrinking of the contacting liquid into a ball shape[30–32]. Similarly, slippery surfaces use low surface tension lubricant layer to separate the liquid from in direct contact with the solid surface[33–35], on which the TCL can recede.

We select three typical substrates, i.e., a fluorinated quartz (F-quartz) substrate, a candle soot-based superamphiphobic substrate, and a lubricant-infused polydimethylsiloxane (S-PDMS) slippery substrate, to investigate the influence of curing interface properties on the one-droplet 3D printing process. All three substrates have high UV transparency, as displayed in Supplementary Fig. 4. The detailed UV curing processes of resin droplets on the F-quartz, superamphiphobic and lubricant-infused PDMS substrates are shown in Fig. 2d–g, j–m, p–s, respectively. All the substrates can facilitate droplet TCL receding. Still, the droplet TCL can recede into a small area only on the superamphiphobic and S-PDMS surfaces during the printing process (Supplementary Movie 1). As Fig. 2b reveals, fluorinating the quartz surface can increase the resin contact angle from 35.1 ± 1.5° to 84.8 ± 1.9° without influencing the surface morphology (Fig. 2c). But the TCL of the resin droplet is pinned on the F-quartz surface during

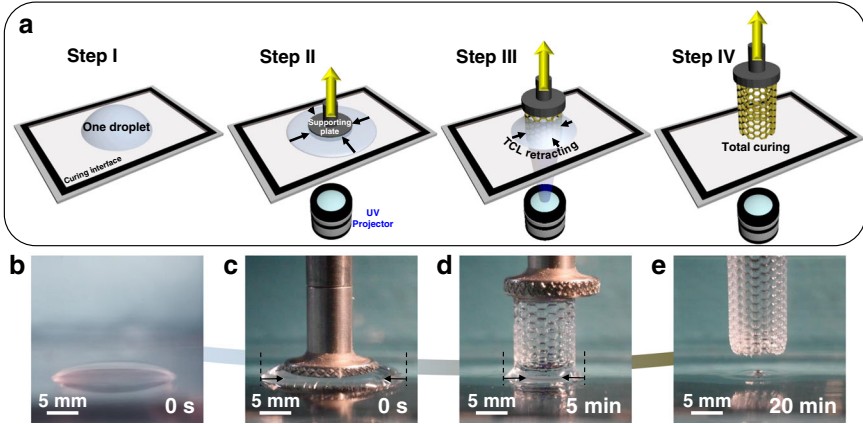

**Fig. 1 Typical schematic and optical images of the one-droplet 3D printing process. a** Scheme of the curing of a single droplet into the desired 3D structure. **b–e** Sequences of optical images of the UV curing-induced TCL receding process for the formation of designed 3D structures from a single droplet. With the continuous projection of the UV illumination pattern and the elevation of the supporting plate, the liquid resin is successively cured into solid structures at the curing interface. Simultaneously, the TCL of the resin droplet recedes with the consumption of the liquid resin. Finally, the liquid droplet is cured into a desired 3D structure.

the continuous printing process (Fig. 2d). Only a short conical structure can be obtained at the curing interface, with the break point at the inner part of the cured structure or at the supporting plate (Fig. 2e). The experimental results achieved on the F-quartz surface are similar to those on the untreated quartz surface (Supplementary Fig. 5). As the liquid resin is in full contact with the F-quartz curing interface, the in situ curing process will transform the adhesion between the resin and the curing interface from pure liquid-solid adhesion to pure solid–solid adhesion (Fig. 2f, g). Large curing-induced adhesion will be generated due to the complete contact mode of the solid–solid interface. The one-droplet printing process cannot be performed on such surfaces.

The superamphiphobic surface used here has a resin contact angle greater than 150° (Fig. 2h) owing to the hierarchical structures and low surface energy (Fig. 2i). A columnar structure can be constructed from the liquid resin droplet during the continuous UV curing process (Fig. 2j). Nevertheless, the sidewall of the columnar structure is not smooth and exhibits striped patterns along the movement direction of the supporting plate (Fig. 2k). The low liquid adhesion property of the super-amphiphobic surface is due to the air entrapment among the hierarchical structures, which results in a composite liquid-air-solid interface (Fig. 2l). A solid-air-solid composite interface is thus formed during UV curing on the superamphiphobic surface, which results in the distortion of the contact line of the resin droplet[36]. And the TCL distortion is solidified and immobilized on the printed structure after UV curing (Fig. 2l), which results in the very low apparent solid–solid adhesion comprised from solid–solid and solid–gas adhesion (Fig. 2m). Therefore, colum-nar structures with vertical stripes on the sidewall are inevitably achieved on the superamphiphobic surface.

For the S-PDMS film, as shown in Fig. 2n, the contact angle of the resin droplet on the curing interface decreases to $42.4 \pm 2.1°$. The infusion of lubricant produces a micro-scale roughness on the surface (Fig. 2o) and a slippery liquid interface between the resin and the solid PDMS[33,37]. Due to the lubricant layer, a composite solid-liquid-solid interface is generated among the cured resin, lubricant, and solid PDMS surface during the UV curing process. Columnar structures with smooth sidewalls can be continuously and stably acquired (Fig. 2p), where the lubricant layer can shield the underlying curing interface from being adhered to by the liquid or solid resin (Fig. 2q). The composite contacting interface between the S-PDMS surface (solid PDMS

and liquid lubricant) and the liquid resin transforms the adhesion to the composite solid–solid and solid-lubricant liquid adhesion during in situ curing process (Fig. 2r, s). In conclusion, comparing with the quartz-based surface with ruptured cured resin and superamphiphobic surface with vertical striped patterns formed on the cured resin sidewall, the S-PDMS surface is thus the best choice for one-droplet 3D printing.

We then investigate the strategic advantage of the one-droplet 3D printing process through comparing with the traditional vat polymerization process using S-PDMS as the curing interface based on two parameters, the time-variant printing width and the ratio ($D/D_{design}$) of the printed width ($D$) to the designed width ($D_{design}$). As displayed in the blue dotted lines in Fig. 2t–u, the printing process is stable with uniform printing widths (left edges range from $-492.9\,\mu m$ to $-507.1\,\mu m$, right edges range from $492.9\,\mu m$ to $500.0\,\mu m$) and stable $D/D_{design}$ ratio (remained between 0.986 and 1.007) for the one-droplet printing process (Supplementary Fig. 6a and Supplementary Movie 2). As both the contact surface between liquid resin and curing interface and the contact surface between liquid resin and air are free ascribing to the TCL receding property on the curing interface (blue lines in Supplementary Fig. 7a, b), it endows a sufficient liquid mobility both at the curing interface and inside the droplet for continuous UV curing. The sufficient inner liquid mobility can also be supported by the vigorous inner liquid flow inside the liquid resin droplet (Supplementary Fig. 8, Supplementary Movie 3), which leads to a stable continuous one-droplet printing process. While for vat polymerization, the printing width becomes nonuniform (left edges range from $-259.1\,\mu m$ to $-663.4\,\mu m$, right edges range from $445.4\,\mu m$ to $704.5\,\mu m$) and the $D/D_{design}$ ratio becomes larger and larger (from 0.981 to the maximum of 1.336 during the 500 s continuous curing process, black dotted lines in Fig. 2t, Supplementary Fig. 6b and Supplementary Movie 2) with the increasing of printing time. It is because that the contact surface between the liquid resin and the curing interface, and the contact surface between the liquid resin and air are confined by the resin vat, which are not free to endow a sufficient curing interface for UV curing (purple lines Supplementary Fig. 7c, d). Protruding or stepping structures on the sidewall, or bubbles generating inside the cured structure by excessive curing will occur at the curing interface during the continuous printing process of vat polymerization (Supplementary Fig. 6b), resulting in the printing instability and the printing precision reduction (black dotted line in Fig. 2u). Besides, the confinement of the

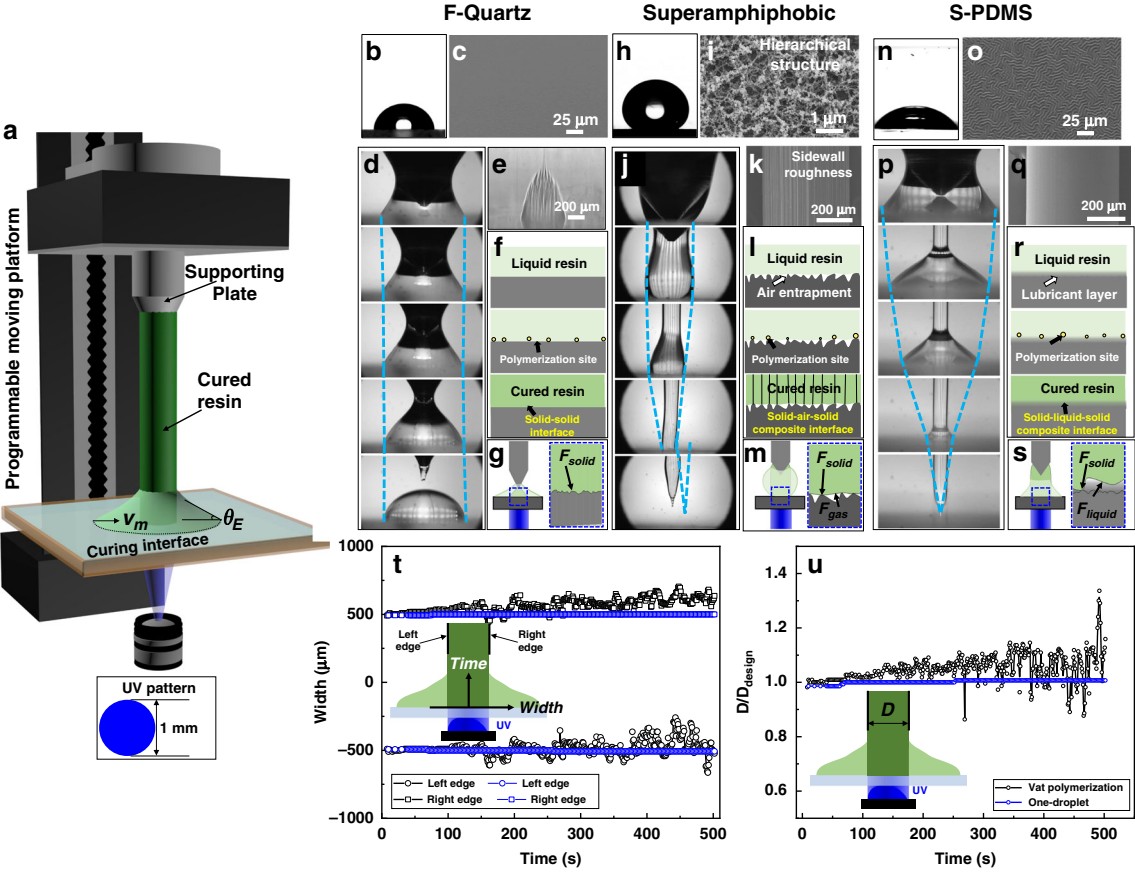

**Fig. 2 The influence of interfacial adhesion properties on the one-droplet 3D printing process. a** Scheme of the experimental setup for the one-droplet 3D printing process. Inset is the scheme of the UV pattern employed in Fig. 2. $v_m$ and $\theta_E$ are the maximum velocity and the equal contact angle on the curing interface for smooth droplet retraction. **b**, **c** Wettability and SEM characterization of the fluorinated quartz (F-quartz) surface, respectively. **d** Optical captures of the UV curing process on the F-quartz surface. **e** Optical image of the cured structure remaining on the F-quartz surface. **f** Scheme of the generation of the pure solid–solid interface on the F-quartz surface. **g** Scheme of the pure solid–solid adhesive force. **h**, **i** Wettability and SEM characterization of the superamphiphobic surface, respectively. **j** Optical captures of the UV curing process on the superamphiphobic surface. **k** SEM image of the sidewall of the cured structure on the superamphiphobic surface, which displays a striped pattern along the printing direction. **l** Scheme of the generation of a solid-air-solid composite interface on the superamphiphobic surface. **m** Scheme of the composite adhesive force comprises solid–solid and solid–gas adhesive force. **n**, **o** Wettability and SEM characterization of the S-PDMS surface, respectively. **p** Optical captures of the UV curing process on the S-PDMS surface to form a columnar structure. **q** SEM image of the smooth sidewall of the cured structure on the S-PDMS surface. **r** Scheme of the generation of a solid-liquid-solid composite interface on the S-PDMS surface. **s** Scheme of the composite adhesive force comprised from solid–solid and solid-lubricant adhesive force. **t** The time-variant width change of the left and right edges on the structure constructed from one-droplet, with vat polymerization process as control. Blue and black lines represent width change curves of the one-droplet and the vat polymerization process, respectively. Circles and squares represent the left and right edges of the printed structure, respectively. **u** The time-variant $D/D_{design}$ ratio of one-droplet printing, with the vat polymerization process as control. Blue and black lines represent $D/D_{design}$ ratio change curves of the one-droplet and the vat polymerization process, respectively.

resin vat can be reflected in the unavailability of the vat resin for curing even with a few printing layers high, *i.e.*, the resin near the cured structure can be completely consumed while the resin near the vat sidewall cannot be used (Supplementary Fig. 9). For printing the same structure through one-droplet 3D printing, TCL can smoothly recede with continuous curing, which leads to the curing of the liquid droplet to a gyroid structure (Supplementary Fig. 10). Therefore, the receding TCL can suppress excessive curing in the x-y plane and improve the printing precision and stability of the continuous 3D printing process.

## Discussion

The influence of the curing interface in controlling the TCL receding behavior as well as the 3D printed structure are further investigated. Based on the experiments demonstrated above, the interfacial adhesions are more critical for one-droplet 3D printing

comparing with the curing interface's contact angle, interfaces involved in the one-droplet printing process and the interfacial adhesion generated at the corresponding interfaces are thus systematically analyzed. As shown in Fig. 3a, three interfaces participate in the one-droplet 3D printing process: (1) the interface between the liquid resin and the cured resin, (2) the interface between the cured resin and the curing interface, and (3) the interface between the liquid resin and the curing interface. The adhesion between the liquid resin and the cured resin ($\gamma_1$) determines the residual amount of liquid resin on the cured structure. It has a constant value for a specific resin system (59.4 $\pm$ 2.9 mJ/m$^2$ of the investigated resin), which can be acquired through the OWRK (Owens, Wendt, Rabel, and Kaelble) method[38–40]. The detailed processes are illustrated in Supplementary Fig. 20 and Supplementary Tables 1–5. The adhesion between the cured resin and the curing interface ($\gamma_2$), i.e., the work needed to separate the cured resin from the curing interface,

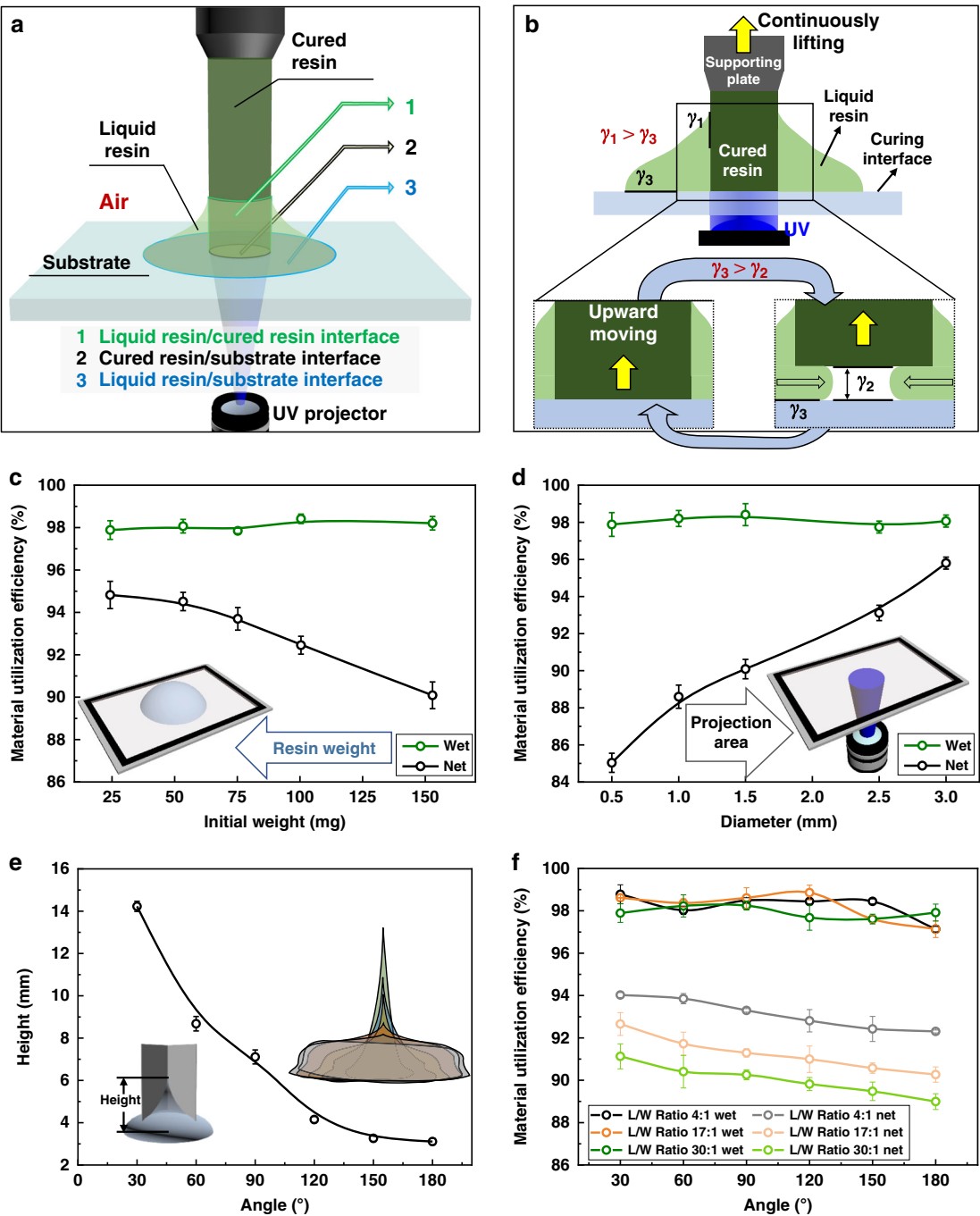

is measured by immobilizing the supporting plate on a load cell[41]. As shown in Table 1, the mean adhesion between the cured resin and the curing interface ($\gamma_2$) is the lowest for the super-amphiphobic surface ($1.5 \pm 0.1$ mJ/m$^2$), followed by the S-PDMS ($1.8 \pm 0.3$ mJ/m$^2$), F-quartz ($67.3 \pm 2.1$ mJ/m$^2$) and quartz ($125.5 \pm 3.4$ mJ/m$^2$) surfaces, as determined by the peel-off test. The adhesion between the liquid resin and the curing surface ($\gamma_3$), which determines the TCL receding process during the one-droplet printing process can also be acquired through the OWRK method. The quartz ($69.4 \pm 1.4$ mJ/m$^2$), F-quartz ($48.8 \pm 3.6$ mJ/m$^2$) and S-PDMS ($56.9 \pm 4.3$ mJ/m$^2$) surfaces possess similar adhesion values, while the superamphiphobic surface possesses a much smaller value ($3.7 \pm 0.6$ mJ/m$^2$).

Based on an analysis of the adhesion of the investigated interfaces, two criteria should be satisfied to ensure a successful one-droplet 3D printing process, as displayed in Fig. 3b. First, the

adhesion between the liquid resin and the cured resin should be greater than the adhesion between the liquid resin and the curing interface, i.e., $\gamma_1 > \gamma_3$ (the upper half of Fig. 3b), so that the TCL can keep receding during the curing-induced liquid resin consumption process, which determines whether the liquid droplet can be cured into the desired 3D structure. Experimentally, $\gamma_1$ is greater than $\gamma_3$ for S-PDMS and superamphiphobic surfaces, on which the droplet TCL keeps receding during the UV curing process. In contrast, the TCL on the quartz surface is pinned, as $\gamma_1$ is less than $\gamma_3$. Second, the adhesion between the liquid resin and the curing interface should be greater than the adhesion between the cured resin and the curing interface, i.e., $\gamma_3 > \gamma_2$ (the bottom half of Fig. 3b). Only then can the cured resin detach from the curing interface to achieve the successive curing process through which continuous 3D printing can occur. Experimentally, $\gamma_3$ is greater than $\gamma_2$ when the S-PDMS or

**Fig. 3 The influence of droplet weight and UV pattern on the resin utilization efficiency. a** Scheme of the three interfaces involved in the one-droplet 3D printing process, including the interface between the liquid resin and the cured resin (1), the interface between the cured resin and the curing interface (2), and the interface between the liquid resin and the solid resin (3). **b** Scheme of the two criteria of one-droplet 3D printing process. **c** The influence of the initial liquid droplet weight on the wet and net material utilization efficiency during the one-droplet printing process. The diameter of the UV pattern is maintained at 1.5 mm, while the initial droplet weight is changed from 24.3 mg to 53.4 mg, 75.2 mg, 100.4 mg and 150.9 mg. Green and black dotted lines represent the wet and net material utilization efficiency, respectively. The error bars in the wet and net material utilization efficiency result from errors in the mass measurements. Each error bar represents the deviation from at least 5 data points. **d** The influence of the UV pattern area on the wet and net resin utilization efficiency in the one-droplet printing process. The weight of the liquid resin droplet is maintained at ~150 mg under the UV projection of a round shape. The UV pattern diameters are 0.5 mm, 1.0 mm, 1.5 mm, 2.5 mm, and 3.0 mm. The error bars in the wet and net material utilization efficiency result from errors in the mass measurements. Each error bar represents the deviation from at least 5 data points. Green and black dotted lines represent the wet and net material utilization efficiency, respectively. **e** Rising height of liquid resin on the V-grooved structures versus the intersection angle of the V-grooved structure (L/W ratio of 17:1). The error bars in the rising height result from errors in the height measurements. Each error bar represents the deviation from at least 5 data points. **f** The influence of the V-grooved angle on the wet and net material utilization efficiency in the one-droplet printing process. Black, orange, and green dotted lines represent the wet material utilization efficiencies of structures with L/W ratios of 4:1, 17:1, and 30:1, respectively. Gray, light orange, and light green dotted lines represent the net material utilization efficiencies of structures with L/W ratios of 4:1, 17:1, and 30:1, respectively. The error bars in the wet and net material utilization efficiency result from errors in the mass measurements. Each error bar represents the deviation from at least 5 data points.

**Table 1 The influence of curing interface property on the interfacial adhesion of the three involved interfaces.**

| Curing interface | Contact angle (°) | Interfacial adhesion (mJ/m²) | | |
| --- | --- | --- | --- | --- |
| | | Cured resin/curing interface ($\gamma_2$) | Liquid resin/curing interface ($\gamma_3$) | Liquid resin/cured resin ($\gamma_1$) |
| Quartz | 34.0 ± 1.3 | 125.5 ± 3.4 | 69.4 ± 1.4 | 59.4 ± 2.9 |
| F-Quartz | 84.8 ± 1.9 | 67.3 ± 2.1 | 48.8 ± 3.6 | |
| S-PDMS | 42.4 ± 2.1 | 1.8 ± 0.3 | 56.9 ± 4.3 | |
| Superamphiphobic | 151.7 ± 2.5 | 1.5 ± 0.1 | 3.7 ± 0.6 | |

superamphiphobic surface is used as the curing interface, with which cured resin can be separated from the curing interface during the printing process. In contrast, the cured resin firmly adheres to the quartz surface, as $\gamma_3$ is less than $\gamma_2$, confirming the second criterion.

For the F-quartz surface, only the first criterion can be satisfied, i.e., $\gamma_1$ is greater than $\gamma_3$. However, as $\gamma_3$ is less than $\gamma_2$, the second criterion cannot be satisfied. One-droplet 3D printing cannot be realized on such a surface, even with a receding TCL. Therefore, the liquid resin droplet can be cured to a predetermined 3D structure only with an appropriate relationship of $\gamma_1 > \gamma_3$ and $\gamma_3 > \gamma_2$ among the interfacial adhesions for the three involved interfaces. The interfacial properties of the S-PDMS are critical as the curing interface for the one-droplet 3D printing process. We further test the one-droplet 3D printing capability of a commercial flexible resin (Flexible Resin, Formlabs, America) on the S-PDMS surface (Supplementary Fig. 11). The relationship of adhesions among the three involved interfaces is consistent with the two criteria mentioned above (Supplementary Tables 2–5), which further proves our conclusion. In addition to the simple and tiny structures, a relatively larger gyroid structure can also be printed through the one-droplet 3D printing strategy on the S-PDMS curing interface (Supplementary Fig. 10).

High material utilization efficiency is an important factor in controlling printing efficiency. Taking S-PDMS as an example, as shown in Fig. 3c–f, all the wet material utilization efficiencies are above 96% for all tested structures ascribing to the receding TCL. Considering the wetting behavior and the capillary phenomenon of the liquid resin on the surface of the cured structure, liquid resin unavoidable adheres to the cured structure surface. Mass loss will occur after the removal of the residual liquid resin on the cured structure. Thus, the weight ratio of the acquired dry structure to the initial liquid resin, which reflects the net material utilization efficiency in the one-droplet printing process, is further investigated. For traditional 3D printing processes performed

in a tank, the adhered amount can be calculated by Bretherton's law[42]. Taking the abovementioned columnar structure as an example, the adhered liquid resin thickness, $e$, scales as $R\text{Ca}^{2/3}$, where $R$ is the radius of the cured column, and Ca is the capillary number which is defined as $\mu U/\gamma$ with $U$ as the elevation speed of the supporting plate. The amount of liquid resin that adheres to the cured part is, therefore, $Q = 2V\text{Ca}^{2/3}$, where $V$ is the volume of the cured resin. For traditional 3D printing processes, the amount of resin adhered to the structures thus remains the same for columns with different radii.

Different from traditional methods, in the one-droplet 3D printing process, the liquid droplet weight (which essentially influences the droplet diameter), projection area of the UV source, and projection pattern of the UV source influence the net material utilization efficiency. This sharp contrast with the traditional 3D printing method performed in a tank results from the pressure drop as the curvature changes from $1/(R + e)$, the curvature of liquid resin adhered to the printed part, to $1/r$, the curvature of the liquid resin droplet. During the printing process, with the transition of the liquid resin to solid resin, the liquid resin droplet on the substrate shrinks from a large size to a small size. When the droplet curvature $1/r > 1/(R + e)$, the adhered resin thickness $e$ scales as $r\text{Ca}^{2/3}$, rather than as $R$. Therefore, a smaller adhered thickness will be obtained at a smaller droplet size. As shown in Fig. 3c, by decreasing the initial weight of the liquid resin droplet while maintaining the UV projection pattern unchanged, the net material utilization efficiency can be increased. Considering the variation point of the dependence of $e$ on the curvature, a larger $R$ means a faster transition of the scaling from $R$ to $r$. As Fig. 3d reveals, with the UV projection pattern set as a round shape and the initial droplet weight unchanged, the net material utilization efficiency increases with the UV pattern radius. Based on a calculation of the surface areas of the structures printed under different experimental conditions (Supplementary Fig. 12a, b), a larger surface area of the 3D

structure leads to a larger amount of residue on the cured structure and a smaller utilization efficiency.

When the UV projection pattern is varied from a round shape to a V-grooved shape, the net material utilization efficiency decreases greatly for traditional printing (Supplementary Fig. 13), but further increases for the one-droplet printing. Variations in the UV pattern will determine the morphology of the printed 3D structure by essentially influencing the contact line morphology and the 3D distribution of liquid resin on the cured structure during the one-droplet 3D printing process. V-grooved samples with intersection angles ranging from 0° to 180° and a fixed surface area of the UV pattern are employed as proof-of-concept examples (Supplementary Fig. 14a, b). During the one-droplet printing process, the resin can be drawn into the V-grooved structure and elevated to a height of $h(x)$, which is the well-known capillary rise effect. The morphology of the contact line changes from a symmetrical round circle for the round shape UV pattern (Supplementary Fig. 15a, b) to an asymmetric shape for the V-grooved shape UV pattern, as shown in Supplementary Figs. 15c, d and 16. As shown in Supplementary Fig. 14c, the V-grooved shape can be considered as two lines intersecting each other, which results in a V-grooved structure with two planes intersecting each other at an angle of $\alpha$. The capillary rise profile of the liquid resin on each plane of the V-grooved structure is essentially a hyperbolic curve, where the highest point is on the intersection line of the two planes, and the lowest position is at the outer sidewall of each plane.

A two-dimensional coordinate system is set as shown in Supplementary Fig. 14c, where the $x-y$ plane is the plane of the symmetric surface of the two planes. The capillary rise height profile $h(x)$ along the symmetric surface of the V-grooved structure at a distance of $x$ can be expressed as[43–45]:

$$h(x) = \frac{2\gamma cos\theta}{\rho\alpha gx} \tag{1}$$

where $\alpha$, $\gamma$, $\rho$, $\theta$ and $g$ are the intersection angle of the V-grooved structure, surface tension of the liquid resin, density of the liquid resin, the contact angle of the liquid resin on the cured resin surface and gravitational constant, respectively.

For the traditional printing process, a large amount of liquid resin store in the V-grooved structures, which greatly reduced the utilization efficiency. However, for the one-droplet printing, with the liquid resin consumed and cured into the V-grooved structure, the capillary force acts as a dragging force to pull the uncured liquid resin to the curing interface. The contact line on the cured structure thus moves downward, opposite to the elevation direction of the supporting plate. The dragging force acts on the liquid resin contact line, significantly influencing the residual amount of liquid resin on the cured structure, and the net material utilization efficiency can be expressed as[27]:

$$F = \frac{W}{vt} = \frac{1}{\alpha} \cdot \frac{4\gamma \cos\theta(1 - \cos\theta)}{\rho gvt} \ln\frac{x_1}{x_0} \tag{2}$$

where $W$ is the work of adhesion of liquid resin on the cured structure, $v$ is the elevation speed of the supporting plate, $t$ is the time needed for printing the entire 3D structure, and $x_1$ and $x_0$ are two positions on the $x$ axis, as shown in Supplementary Fig. 14c (the detailed derivation process can be found in the Supplementary Information).

For a predetermined system, $\gamma$, $\rho$, $\theta$, $g$, $v$, and $t$ are all constant values. Therefore, the capillary force is inversely proportional to the intersection angle $\alpha$ and the distance $x$. Macroscopically, for a specific value of $x$, a smaller intersection angle leads to a higher elevation height $h(x)$ of liquid resin on the structure (Fig. 3e and Supplementary Fig. 15), thus leading to a much larger capillary force and a lower amount of liquid resin residue on the cured

structure. A higher net material utilization efficiency can thus be achieved with a smaller intersection angle, which is consistent with the experimental results shown in Fig. 3f. Varying the length-to-width ratio (L/W ratio) while keeping the intersection angle and surface area of the UV pattern unchanged, a larger material utilization efficiency is acquired at a smaller L/W ratio (Fig. 3f and Supplementary Fig. 17), which is consistent with the result shown in Fig. 3c, d. Thus, the net material utilization efficiency is related to the surface area of the 3D structure (Supplementary Fig. 12c) and the 3D structure-related capillary force on the interface between the liquid resin and the cured resin.

We further compare the results with the wet and net material utilization efficiency in commercial 3D printing based on UV curing. As shown in Supplementary Fig. 13, 10.18 g of resin is added to a homemade resin vat, the residue is weighed, and the corresponding material utilization efficiency is calculated. Both the wet material utilization efficiency (~87.5%) and the net material utilization efficiency (~66.7%) are much lower than those from the one-droplet 3D printing process (~98.6% for wet and 92.5% for net material efficiency, Fig. 3f, L/W ratio of 17:1, 30°). As shown in Supplementary Fig. 18, the one-droplet 3D printing process with higher material utilization efficiencies can print sharper V-grooved structures than vat polymerization, which can be due to the dragging force (Eq. 2) acting on the liquid resin receding contact line, where extra curing due to residual can be partly suppressed comparing with vat polymerization. Therefore, one-droplet 3D printing process with higher material utilization efficiencies not only can reduce the residual, but also can increase the printing precision.

Encouraged by the increased material utilization efficiency, we further demonstrate the capability of the one-droplet 3D printing process in printing a tooth structure, which will be useful for personalized customization of dental investigations (Fig. 4a and Supplementary Fig. 19). The tooth model can be generated from a 3D scanning file obtained with medical micro-computed tomography[46]. After slicing the source file into layer images, the tooth structure can be 3D printed from a single droplet through the continuous projection of the layer images. As shown in Fig. 4b, d, with continuous projection of sequential layers of UV patterns on the curing interface, the tooth structure is continuously cured while the TCL of the resin droplet keeps receding on the curing interface. Finally, a liquid resin droplet is completely cured into a single tooth structure (Supplementary Movie 4). To illustrate the structural controllability of this method, molar, incisor, and canine teeth crown structures are prepared as representative examples (Fig. 4e), for which three individual resin droplets are employed (Fig. 4f). The printed molar, incisor and canine crown structures have smooth sidewalls and are suitable for dental treatment (Fig. 4g). As shown in Fig. 4h, the inserted molar, incisor and canine teeth crown structures display appropriate contact with adjacent teeth and marginal integrity. In addition, the three-dimensional dewetting process of liquid resin on the cured tooth structures with different curvatures in different regions can be visualized, which provides a strategy to monitor the 3D dewetting process. Thus, one-droplet 3D printing can completely transform a liquid resin droplet into a defined structure with high material utilization efficiency through proper adjustment of the curing interface properties, which can save valuable inks and provide reference for on-demand 3D printing.

In summary, the concept of interfacial property regulation enables the receding of the resin droplet TCL during the UV curing process, which leads to the efficient curing of droplet to a desired 3D structure. The 3D distribution and the dewetting force of liquid resin on the cured structure can be well controlled by regulating the droplet dimension and UV pattern parameters,

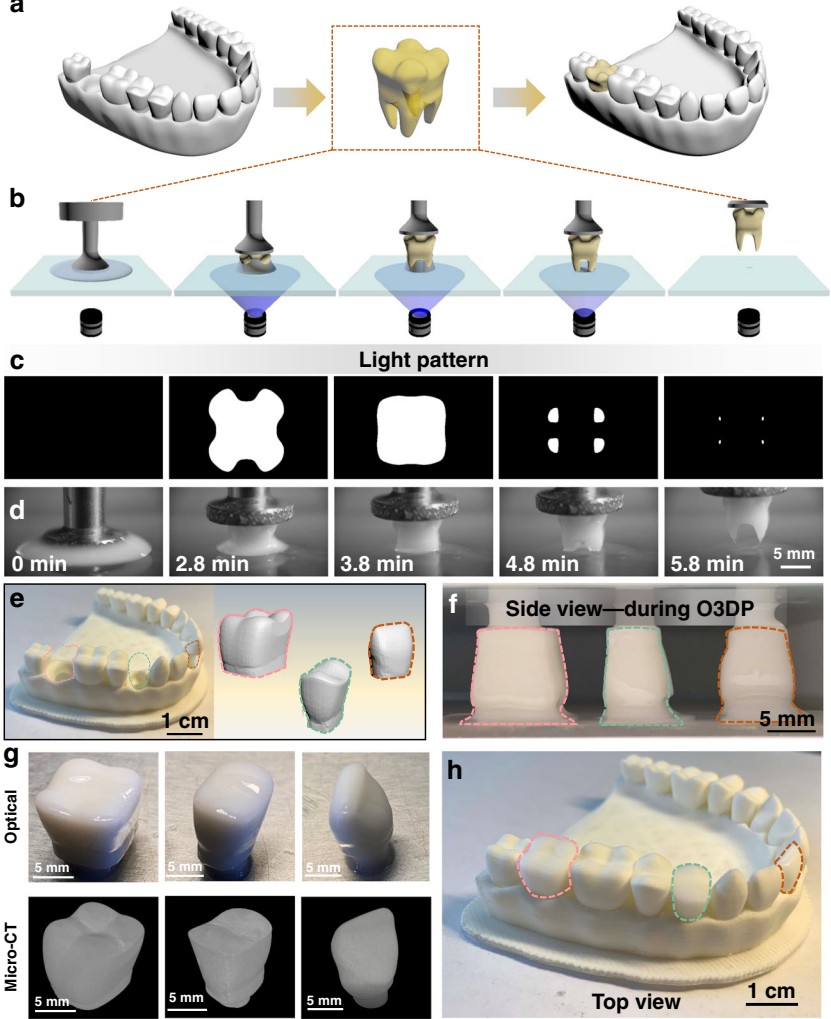

**Fig. 4 Fine tooth structure fabricated through the one-droplet 3D printing process. a** Scheme of the generation of the tooth model for the one-droplet 3D printing process. **b**, **c** Schemes (**b**), UV pattern sequences (**c**) and corresponding optical images (**d**) of the one-droplet 3D printing process, in which a resin droplet is cured into a desired tooth structure. **e** Scheme of the three selected crown structures and corresponding opposing dentition. Pink, green and orange indicate the crowns of a molar, a canine and an incisor. **f** Optical image of the one-droplet 3D printing process for printing artificial crown structures. **g** Optical and micro-CT characterization of the printed crown structures, including the printed molar, canine, and incisor crowns structures. **h** Fixation of the printed crowns into the model with opposing dentition, with appropriate contact with adjacent teeth and marginal integrity.

resulting in minimization of liquid resin residue and a high material utilization efficiency. Extra curing with protruding and stepping sidewalls can be prevented due to the free contact surface property of the droplet system. With the receding TCL, the inner droplet liquid circulation is increased and the adhesion properties among the liquid resin, cured resin, and resin vat are reduced. This strategy to construct fine 3D structures from a single droplet with high efficiency will be of great significance for on-demand 3D fabrication.

## Methods

**Resin preparation**. A UV curable resin system was prepared through mixing prepolymer, reactive dilute, photo initiator, and other additions. Here, polyacrylate system is employed and prepared from prepolymer acrylic resin, monomer di (ethylene glycol) ethyletheracrylate, photo initiator 2,4,6-phenylbis (2,4,6-tri-methylbenzoyl) phosphine oxide and crosslinker poly(ethylene glycol) diacrylate 700.

**Quartz surface**. The quartz plate was cleaned with acetone, ethanol, and deionized water before use.

**Fluorinated quartz surface**. The fluorinated quartz surface was prepared by the fluorination of the quartz plate. The quartz plate was firstly treated with $O_2$ plasma (under 70 Pa, 100 mW for 200 s). Then it was put into a vacuum dryer, to silanize the quartz surface with 1H,1H,2H,2H-perfluorodecyltrimethoxysilane (PFOS) in a decompression environment at 80 °C for 3 h. After taking out from the dryer, the fluorinated quartz surface is obtained.

**Lubricant-infused PDMS surface**. The PDMS pre-polymer and curing agent (Sylgard-184 silicone elastomer, Dow Corning, USA) were mixed with a ratio of 10:1, stirred by a mechanical stirrer for 10 min, and poured onto a clean quartz surface. Then it was put in a vacuum oven to remove bubbles. After putting into an oven heated at 60 °C for 4 h. Through immersing the PDMS surface in perfluoro-carbon for 24 h, the S-PDMS surface was acquired. The lubricant-infused surface is stored by covering with a thin layer of lubricant.

**Superamphiphobic surface**. The superamphiphobic surface is prepared through the superamphiphobic modification of the quartz surface. Placing the cleaned quartz plates above the flame to deposit a layer of soot. The soot coated quartz plates were placed in a vacuum dryer together with two small open beaker containing 1 ml of tetraethoxysilane (TES) and aqueous ammonia solution, respectively. Then chemical vapor deposition of TES was carried out for 48 h. Similar to a Stöber reaction, silica is formed by hydrolysis and condensation of TES. After calcination at 600 °C for 4 h in air, treated by $O_2$ plasma at 150 mW for 300 s, and

immersed into (heptadecafluoro-1,1,2,2-tetradecyl) trimethoxysilane solution in hexane (1 mg/mL) for 2 h, the superamphiphobic surface is acquired.

**3D printing apparatus**. The 3D printing apparatus is self-made with the composition of LED UV projector (PRO4500, Wintech, China), liquid resin vat (self-made), aluminum supporting plate (self-made) mounting on a programmable moving platform (MC600, Zolix Instruments Co., Ltd. China) from bottom to up as displayed in Fig. 2a. The UV projector can provide light patterns with a projection area of 32.2 mm × 51.6 mm, resolution of 912 × 1140 pixels and light intensity range 0–65 mW/cm². The moving velocity range of the programmable moving platform is 1.5–100 mm/min with a resolution of 10 μm.

**Characterization**. SEM images were obtained using a field-emission scanning electron microscope (SEM, JSM-7500F, JEOL, Japan). Internal structure characterization was shot and reconstructed through a microcomputed tomography (Micro-CT) equipment (Skyscan 1272, Bruker, Germany). Contact angles were measured with a contact angle measurement equipment (OCA20, DataPhysics, Germany) with droplets of 3.0 μL. Each reported contact angle was an average of at least five independent measurements. The surface tensions of the liquid resins were measured through a high-sensitivity microelectromechanical balance system (DataPhysics DCAT 11, Germany). Real-time monitoring of the UV curing process on different curing interfaces is conducted through a digital camera (D750, Nikon, Japan). The adhesive force was measured using the digital load cell (M5-05, Mark-10 Corporation, America), which was mounted on the programmable moving platform (ESM303, Mark-10 Corporation, America). The threshold force for the load cell is 2.5 N with a force resolution of 0.5 mN and a displacement resolution of 20.0 μm. Force versus distance data was collected by the load cell while supporting plate moving.

## Data availability

The authors declare that the main data supporting the findings of this study are contained within the paper. Source data relating to Figs. 2t, u and 3c–f are provided as an additional file published with their work. All other relevant data are available from the corresponding author upon reasonable request. Source data are provided with this paper.

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

## Acknowledgements

We acknowledge funding of the National Key R&D Program of China (Grant Nos. 2018YFA0208501, 2016YFB0401603, 2016YFC1100502, and 2016YFB0401100), the National Natural Science Foundation (Grant Nos. 51803219, 51773206, and 91963212), K.C. Wong Education Foundation and Beijing National Laboratory for Molecular Sciences (Grant No. BNLMS-CXXM-202005). N. X. Fang and H. Du are grateful for the seed provided by the MIT Energy Initiative.

## Author contributions

L.W. and Y.S. conceived and designed the experiments., Y.Z., Z.D., C.L., H.D., and L.W. performed the experiments. Y.S. and L.W. analyzed the data. L.W. wrote the original paper, Y.S. and N.F. helped revise it. All authors discussed the results and commented on the paper.

## Competing interests

The authors declare no competing interests.
