## [Peer Review File · Nature Communications]

REVIEWER COMMENTS

Reviewer #1 (Remarks to the Author):

This paper describes a new VAT polymerisation process capable of printing continuously from a single droplet of resin. The aim is to reduce material waste which is partly realised by controlling the three-phase contact line (TCL). The paper is generally of good quality.

The main claims is the reduced resin waste (high material utilization efficiency) by printing from a single droplet.

Comments:

- 1) It is not clear from the paper why an improved material utilization efficiency is of particular importance. The resin for VAT polymerization processes can be reused with the exception of trapped/adhered resin on the part before rinsing. As the process is not scalable the interest for the community will be limited to niche applications and not have the broader reach expected for Nature Communication.
- 2) P12L11: The claim that the proposed process can suppress excessive curing in the x-y plane which implies a better resolution is not convincingly evidenced.
- 3) The paper convincingly proves that the one-droplet process reduces material waste for certain geometries. The advantage does however seem to be entirely geometry dependent and it should be made clearer if the proposed process is always an advantage.
- 4) The experimental setup not described in the manuscript.
- 5) Too much information is "hidden" in supplementary information and should be moved to the main manuscript if deemed essential.
- 6) The discussion on contact angles pp 7-8 becomes very confusing without a figure to support the descriptions.

Reviewer #2 (Remarks to the Author):

In this manuscript, the authors develop a novel strategy for improving the material utilization efficiency. The utilization efficiency of the materials is of importance for 3D printing. Currently, the common methods to improve the material efficiency mainly focus on optimizing the geometry structure or enhancing the printing processes. Here the authors provide an alternative method that is very interesting, simple and useful. The authors proposed an interesting concept by a one-droplet 3D printing process. Through focusing on the interfacial adhesion, the curing interfaces property and the droplet TCL shrinking process can be well controlled. As they mentioned, this method has achieved the record material utilization efficiency and high printing speed. This research is very useful for practical applications. Therefore, the reviewer would like to recommend the acceptance of

this manuscript for publication in Nature Communications after minor revision suggested below.

1. Can UV resin with different surface tensions be versatile to this method?
2. Would the lubricant leak from the curing interface to the printed structure? Then how to keep the lubricating liquid should be described in detailed.
3. The influence of contact angle and adhesive energy on the 3D printing is not clear, the authors should explain it in the manuscript.
4. Is the candle soot-based hierarchical surface optically transparent? The optical characterization of the candle soot-based hierarchical surface should be supplemented.
5. The reviewer is a little bit confused about whether some liquid resin is left on the supporting plate? As the supporting plate should be high adhesive to ensure the printing of the first layer.
6. Several abbreviations need to be spelled out before using them (For example, OWRK on P. 7, UV on P. 1)

Reviewer #3 (Remarks to the Author):

The paper presents an interesting approach to a Vat Polymerization process. The main claims are related to material utilization improvement and cost reduction.

However, it can be expected that these aspects are less important for products of the size associated with this process type. It is also stated that an improved level of detail can be expected, which might be of greater importance for small (medical) artifacts. It would strengthen the paper if the claims in the direction of improvements for the end-user (especially costs and accuracy improvement) for a typical product (for example the tooth) are supported by numbers.

Detailed comments, supporting the above, are included in the pdf.

T. Vaneker

Responses to Reviewer # 1

This paper describes a new VAT polymerisation process capable of printing continuously from a single droplet of resin. The aim is to reduce material waste which is partly realised by controlling the three-phase contact line (TCL). The paper is generally of good quality. The main claims is the reduced resin waste (high material utilization efficiency) by printing from a single droplet.

Reply: We greatly appreciate the reviewer for the positive assessment and valuable suggestions. According to the reviewer's comments, the manuscript has been carefully revised. We hope that the revised manuscript would be suitable for publication in *Nature Communications*.

1. It is not clear from the paper why an improved material utilization efficiency is of particular importance. The resin for VAT polymerization processes can be reused with the exception of trapped/adhered resin on the part before rinsing. As the process is not scalable the interest for the community will be limited to niche applications and not have the broader reached expected for Nature Communication.

Reply: We thank the reviewer very much for the comment. We are sorry for the omission of the comparison with the vat polymerization process. To directly display the advantage of one-droplet 3D printing, we have conducted additional experiments involving the comparison with vat polymerization to distinguish the priority of the one-droplet 3D printing strategy. In brief, the continuous vat polymerization process not only pollutes or cures the resin non-quantitatively but also affects the printing precision. Considering the printing speed, precision, resin utilization efficiency are all essential factors for 3D printing, the one-droplet 3D printing method is important for readers with various backgrounds.

As a prerequisite, we want to emphasize that continuous 3D printing is different from intermittent 3D printing process (including intermittent vat polymerization and intermittent one-droplet printing processes). During intermittent 3D printing, UV is on when curing, and is off during the interval. Traditionally, the interval is employed for cured resin separation, heat dissipation inside the liquid resin, and the resin refilling between the cured resin and the curing interface for the generation of the next curing layer. While during continuous 3D printing, UV is always on with continuous projection of UV patterns onto the curing interface during the whole process, which leads to the continuous irradiation-induced accumulation of energy at the curing interface inside the liquid resin. In addition to the requirement of sufficient heat dissipation, the requirement of resin mobility, which influences the generation of a new curing interface, is also more critical for continuous printing than intermittent printing. All experiments are conducted without damaging the curing interface, *i.e.*, the results are concluded with the exclusion of the influence of the curing interface.

Figure R1 (Supplementary Figure 1 in the revised Supplementary Information). Afterglow of the “dark” region of the UV projector under different UV intensities. **a.** Scheme of the patterned region and the non-patterned region of the UV pattern projected from the UV source. The star shape is the tested pattern and the white frame represents the breath of the UV projector. **b - i.** Optical images of the UV pattern when projecting the pattern in **a** with different UV intensities (from 1 mW/cm² to 40 mW/cm²). The images are captured with the camera settings (including the exposure time, magnification, photosensitivity and aperture, *et al.*) unchanged. The UV projector can provide 405 nm pattern with the projection area of 51.6 mm × 32.2 mm. With the increasing of the light intensity, the afterglow across the whole non-patterned dark region increases significantly, which finally may lead to the curing of the non-patterned dark region.

First, the afterglow of the “dark” region of the UV pattern will pollute or even cured the vat resin. Traditionally, a slice of UV pattern contains the patterned region and the non-patterned region (Figure R1a), where the UV patterned region displayed light of predetermined intensity, and the non-patterned region should be completely dark or 0 mW/cm². However, current DLP light source is based on LED. The “dark” region, *i.e.*, the non-patterned region of a projected slice, is not really 100% “black” or 0 mW/cm² UV intensity. As shown in Figure R1b - 1i, with the increasing of the UV intensity, the afterglow of the “dark” region becomes brighter and brighter. Just as we mentioned above, the continuous printing process requires the continuous projection of the UV light with a certain intensity. The afterglow of the “dark” region will lead to the unintentional generation of free radicals and result in the inhomogeneous composition of the vat resin. The amount of free radical is not generated quantitatively, and the worst condition is the trigger of the curing of the un-patterned “dark” region.

Figure R2 (Supplementary Figure 2 in the revised manuscript). The unintentional curing of the “dark” region during the vat polymerization process. **a.** Optical image of the vat after continuous printing the typical pattern (1 mm round shape) characterized in the Figure R7 (UV intensity of 30 mW/cm^2 and printing speed of $100 \text{ }\mu\text{m/s}$). The blue dot and the square black dotted frame are scheme of the UV pattern and the breadth of the UV projector, respectively. **b.** Optical image of the unintentionally cured resin. **c-e.** Series of optical captures of the curing of the “dark” region which influences the printing precision. Insets are enlarged view of corresponding regions. Red circles indicate the cured part of the “dark” region which connects to the patterned region.

As shown in Figure R2, after continuous printing the typical pattern (1 mm round shape with UV intensity of 30 mW/cm^2 and printing speed of $100 \text{ }\mu\text{m/s}$) used in the manuscript, the non-patterned region is unintentionally cured, which cannot be easily recognized without leaning the vat. Besides, for printing 3D structures with a large area of UV patterned regions, the cured part by the afterglow may be connected with the UV patterned region (Figure R2c-e, Movie R1 which real-time monitors the curing of the “dark” region which connects to the patterned region), which will further influence the printing precision. Therefore, for continuous 3D printing process, the continuous projection of the UV light will pollute the resin non-quantitatively and will affect the printing precision.

Figure R3 (Supplementary Figure 3 in the revised Supplementary Information, Movie R2). Optical captures of the cured structure after repeatedly using the same vat resin through vat polymerization. The experiment is

conducted through repeatedly utilizing the same vat liquid resin at the same position with the same UV pattern, UV intensity and printing speed (50 mW/cm^2 , $200 \text{ }\mu\text{m/s}$).

Second, we have found that the repeating use of the vat resin will decrease the 3D printing precision. We have repeatedly used the same location on the curing interface to continuously print the same structure from the same vat resin for five times (50 mW/cm^2 , $200 \text{ }\mu\text{m/s}$). For the first time, the newly added liquid resin can print a relatively uniform structure. With the increase of the repeating time, the uniformity of the printed structure decreased, which leads to rough protruding on the structure sidewall (yellow circles in Figure R3, Movie R2 which real-time monitors the repeated utilization of vat resin during the vat polymerization process). Therefore, our proposed one-droplet 3D printing strategy not only avoids the reutilization of vat resin but also can solve this problem intrinsically.

Figure R4 (Supplementary Figure 9 in the revised Supplementary Information). The unavailability of vat resin for curing even with a few printing layers high. **a-e**. Optical captures of the vat polymerization process of the gyroid structure with the dimension in the x-y plane of $4.5 \text{ cm} \times 2.3 \text{ cm}$ (5 mW/cm^2 , printing speed of $20 \text{ }\mu\text{m/s}$). The red lines in **a-d** are outlines of the upper surface of the vat resin.

Third, the vat resin is confined by the vat sidewall and the vat bottom, which leads to the confined liquid mobility and the unavailability of liquid resin as resin source for curing even with a few printing layers high and under a low printing speed. We have compared the liquid resin mobility of vat polymerization with one-droplet printing process under the condition when the amount of vat resin is theoretically available to print the designed 3D structure.

As displayed in Figure R4, the TCL of the vat resin is pinned on the sidewall and the bottom of the vat. With the continuous consumption of the vat resin, the upper surface of the liquid resin distorted with the upper surface of the liquid resin near the sidewall higher than the inner surface, as displayed by the red dotted lines in Figure R4a - d. Finally, the resin near the cured structure is completely consumed, while the resin near the vat sidewall cannot be utilized, which leads to the failure of further printing. Therefore, the liquid remains inside the vat with certain resin thickness because of the confinement of the contact surface by the vat, so that the vat resin cannot be utilized even with a few printing layers high.

Figure R5 (Supplementary Figure 10 in the revised Supplementary Information). One-droplet 3D printing a gyroid structure. **a-e.** Optical captures of the one-droplet printing process of the gyroid structure with the dimension in the x-y plane of 4.5 cm × 2.3 cm. **f.** Optical capture of the printed structure after UV off. **g.** Optical image of the supporting plate employed for printing the gyroid structure. **h.** Optical image of the gyroid structure printed from one-droplet.

While for one-droplet 3D printing of the same structure, as displayed in Figure R5, the TCL of the sizeable liquid resin droplet can continuously recede along with the continuous printing process, which proves the freer contact surfaces or higher TCL mobility of the one-droplet printing process. Therefore, the resin mobility, *i.e.*, the regeneration speed of a new curing interface, is high for one-droplet 3D printing process to maintain a continuous printing process.

Figure R6 (Supplementary Figure 8 in the revised Supplementary Information). Optical tracking of the inner liquid flow during the continuous one-droplet 3D printing process through adding lightweight and black carbon nanofibers inside the liquid resin droplet. **a.** The initial position of Particle A (red circle) in the liquid resin droplet. **b.** The initial position of Particle B and part of the trajectory of particle A. Blue circle represent the initial position of Particle B. **c.** The trajectories of particle A and particle B in the liquid resin and cured resin structure. Red and blue dot lines indicate the trajectories of particle A and particle B, respectively. Red dot lines indicate the trajectories of particle A and particle B, respectively. **d.** The trajectories of particle A and particle B in the liquid resin and cured resin structure. Red and blue dot lines indicate the trajectories of particle A and particle B, respectively.

Fourth, the liquid motion inside the droplet and at the curing interface is vigorous during the continuous printing process. We have added immiscible additive inside the transparent resin droplet to track the internal liquid flow inside the droplet during the continuous printing process. Carbon nanofibers are selected as the additive, as their color is black for easy tracking, and their weight is light enough to flow along with the internal liquid flow easily. As shown in Figure R6 (Supplementary Movie 3), with the continuous lifting of the supporting plate, the inner droplet liquid can flow along the liquid resin-air interface upward to the cured structure-liquid resin interface (the trajectory of Particle A in Figure R6a-c) driven by the adhesion of γ_1 , or along the sidewall of the cured structure downward to the curing interface (the trajectory of Particle A in Figure R6c-d and the trajectory of Particle B in Figure R6c) driven by the adhesion of γ_3 . The liquid resin inside the whole droplet is thus continuously moving, and the liquid resin is mixed uniformly during the continuous printing process.

Therefore, as the employed liquid resin droplet is cured to a designed 3D structure, the one-droplet 3D printing method avoids the repeatable irradiation of liquid resin with UV light, neither by the UV pattern nor by the afterglow of the “dark” region. Moreover, the receding TCL endows higher liquid mobility both inside the droplet and at the curing interface during the continuous printing process. In reply to the 2nd comment of the reviewer, we have experimentally and quantitatively evidenced the improved printing precision of the one-droplet printing process (Figure R7 and Figure R8) through comparing with vat polymerization, where extra curing during continuous printing process can be prevented. In addition, the high material utilization efficiency of the one-droplet 3D printing process can result in sharper V-grooved structure as shown in Figure R9 (in response to the 3rd comment of the reviewer), which further proves the strategic advantage of one-droplet 3D printing.

Finally, the essential advantage of 3D printing is the on-demand manufacture capability without utilizing molds, which avoids inevitable wastes. Current investigations have made intensive improvement on the 3D printing speed, which makes 3D printing more applicable than before. For the next stage of 3D printing applications, the material utilization efficiency, which really reflects the advantage of 3D printing should be on the investigation agenda.

Accordingly, we have revised the **Introduction** part in the revised manuscript, and added new experimental results in the revised Supplementary Information.

2. P12L11: The claim that the proposed process can suppress excessive curing in the x-y plane which implies a better resolution is not convincingly evidenced.

Reply: Thanks for the reviewer’s comment. According to the reviewer’s suggestion, we directly monitor the *in situ* curing process of vat polymerization and one-droplet printing process in real-time with the same experimental conditions to quantitatively compare the printing precision and evidence the improved precision. Two parameters, including the printing width of the left and right edges which reflects the outer profile of the printed structure, and the ratio (D/D_{design}) of the printed width (D) to the designed width (D_{design}) which reflects the extra curing phenomenon, are characterized for comparison.

Figure R7 (Supplementary Figure 6a-b in the revised Supplementary Information and Figure 2h-i in the revised manuscript). Comparison of the continuous printing process of the vat polymerization and the one-droplet

3D printing process (printing speed of 100 $\mu\text{m/s}$, 500 s duration for printing structure of 5 cm high). **a.** Series of optical captures of the one-droplet 3D printing process. **b.** Series of optical captures of the vat polymerization process. **c.** The time-variant width change of the left and right edges on the structure constructed from one-droplet continuous printing, with the vat polymerization process as control. Red and black lines represent the width change curves of the one-droplet printing process and the vat polymerization process, respectively. Circles and squares represent the left and right edges of the printed structure, respectively. **d.** The time-variant D/D_{design} ratio change of the one-droplet printing process, with the vat polymerization process as control. Red and black lines represent D/D_{design} ratio change curves of the one-droplet printing process and the vat polymerization process, respectively. Scale bars are 1 mm.

As displayed in Figure R7a and the red dotted lines in Figure R7c, the printing process is stable with the left and right edges keeping straight during the whole process (left edges range from $-492.9 \mu\text{m}$ to $-507.1 \mu\text{m}$, right edges range from $492.9 \mu\text{m}$ to $500.0 \mu\text{m}$), as well as the stable D/D_{design} ratio (remained between 0.986 and 1.007, the red dotted line in Figure R7d). While for vat polymerization, the printing width becomes nonuniform (left edges range from $-259.1 \mu\text{m}$ to $-663.4 \mu\text{m}$, right edges range from $445.4 \mu\text{m}$ to $704.5 \mu\text{m}$) with protruding or stepped structures on the sidewall along with the increasing of printing time (Figure R7b and the black dotted lines in Figure R7c). The D/D_{design} ratio also becomes larger (from 0.981 to the maximum of 1.336 during the 500 s continuous curing process) and unstable with the increasing of the time (black dotted line in Figure R7d). The real-time characterization of the two processes is displayed in Supplementary Movie 2.

Figure R8 (Supplementary Figure 7 in the revised Supplementary Information). Comparison of the contact surface mobility of the one-droplet 3D printing and the vat polymerization process. **a.** Scheme of the configuration of the one-droplet 3D printing process. **b.** Scheme of the free contact surfaces during the one-droplet 3D printing process. Blue lines are the outlines of the free contact surfaces. **c.** Scheme of the configuration of the vat polymerization process. **d.** Scheme of the confined contact surfaces of the vat polymerization process. Purple lines are the confined contact surfaces.

The difference in the printing process and the printing precision can be due to the difference in the confinement effect of the contact surfaces. For the one-droplet 3D printing process, the liquid resin can recede during the curing process (Figure R8a-b). The contact surface between the liquid resin and the curing interface, and the contact surface between the liquid resin and the air are free. It endows sufficient resin mobility on the curing interface for continuous UV curing. Therefore, the one-droplet printing process is dominated by the relationship among the three interfaces, as described in Figure 3a-b in the revised manuscript. The sufficient inner liquid mobility can also be supported by the vigorous inner liquid flow inside the liquid resin droplet (Figure R6), which leads to a stable and continuous one-droplet printing process.

In contrast, for vat polymerization, the contact surface between the liquid resin and the curing interface, and the contact surface between the liquid resin and the air are confined by the vat. Under this confinement, the vat resin prefers to be cured rather than be freely moving to endow a sufficient curing interface for UV curing (Figure R8c-d). Extra curing with protruding or stepped structures on the sidewall or with bubbles generating inside the cured resin occurs during the continuous printing process (Figure R7b), which results in printing instability and decrease in printing precision (black dotted line in Figure R7d). Besides, the confinement of the resin by the vat can also be reflected in the unavailability of the vat resin to serve as resin source for curing even with a few printing layers high, but with the complete consumption of the resin near the cured structure while higher resin upper surface near the vat sidewall (Figure R4). And the availability of one-droplet strategy to print the same structure (Figure R5). Therefore, the confinement of the liquid resin to a droplet morphology with receding TCL can suppress excessive curing in the x-y plane, and improve the printing accuracy and stability of the continuous 3D printing process.

Accordingly, we have added corresponding experimental results as Figure 2h-i in the manuscript, Supplementary Figure 6-7 in the revised Supplementary Information and Supplementary Movie 2 to evidence the better resolution of one-droplet 3D printing convincingly.

3. The paper convincingly proves that the one-droplet process reduces material waste for certain geometries. The advantage does however seem to be entirely geometry dependent and it should be made clearer if the proposed process is always an advantage.

Reply: Thanks for the reviewer's comment. In the manuscript, we have investigated two material utilization efficiencies, the wet material utilization efficiency, and the net material utilization efficiency. We are sorry for the missing of the wet material utilization efficiencies in Figure 3, and we have added them in the revised Figure 3. The wet material utilization efficiency is geometry independent, while the net material utilization efficiency is geometry dependent. The high wet material utilization efficiency is always an advantage, as we discussed in the 1st and 2nd comments above. The lower net material utilization efficiency than the wet material utilization efficiency is intrinsically due to the adhesion between the cured structure and the liquid resin. Theoretically, the residual on the cured structure should be as small as possible, as the cured structure is continuous under the UV irradiation. The refracted or reflected light from the UV pattern or the afterglow of the "dark" region (as displayed in Figure R1, Figure R4 and Figure R5) to the cured structure will

potentially cure the adhered residual and decrease the printing precision. Imagine the extreme situation, if there exists a kind of resin that the liquid monomer does not adhere to the cured resin at all, the situation mentioned above will not happen, and post-washing is even no more in need as well, which really realizes the 100% on-demand 3D printing. Thus, we think the net material utilization efficiency should also be as high as possible, where the liquid resin can be on-demand utilized for printing 3D structures.

Figure R9 (Supplementary Figure 18 in the revised Supplementary Information). Micro-CT characterization of the V-grooved structures prepared from vat polymerization and one-droplet 3D printing. **a**, **b** and **c** are inner side view of the designed, Micro-CT image of V-grooved structure prepared from vat polymerization and Micro-CT image of V-grooved structure prepared from one-droplet 3D printing, respectively. **d**, **e** and **f** are outer side view of the designed, Micro-CT image of V-grooved structure prepared from vat polymerization and Micro-CT image of V-grooved structure prepared from one-droplet 3D printing, respectively. **g**, **h** and **i** are cross-sectional view of the designed, Micro-CT image of V-grooved structure prepared from vat polymerization and Micro-CT image of V-grooved structure prepared from one-droplet 3D printing, respectively. The blue V patterns in **h** and **i** are the designed V patterns. Scale bars are 2 mm.

The reviewer concerned whether the increase of the net material utilization efficiency is an advantage. According to the reviewer's comment, we have compared the detailed morphologies of

the asymmetric V-groove structures (with intersection angle of 30°) printed from one-droplet 3D printing (wet material utilization of ~98.6% and net material utilization efficiency of ~92.5%, Figure 3f in the revised manuscript) and from vat polymerization (wet material utilization of ~87.5% and net material utilization efficiency of ~66.7%, Supplementary Figure 13) through Micro-CT characterization. As shown in Figure R9, one-droplet 3D printing process can print sharper inner V-corner than vat polymerization. This can be due to the dragging force (Equation 2 in the manuscript) acting on the liquid resin contact line, which leads to higher net material utilization efficiency and less residual inside the V-corner. However, the residual on the cured structure and cannot be avoided entirely due to the similar compatibility principle, as the compositions and functional groups are similar for monomers and cured monomers. In other words, extra curing due to residual at the inner V-corner can be partly suppressed during one-droplet 3D printing process. Therefore, one-droplet 3D printing process with higher material utilization efficiencies not only can reduce the residual, but also can increase the printing precision, which is also an advantage.

Accordingly, we have added the experimental data as Supplementary Figure 18 in the revised Supplementary Information with corresponding descriptions: “As shown in Supplementary Figure 18, the one-droplet 3D printing process with higher material utilization efficiencies can print sharper V-grooved structures than vat polymerization, which can be due to the dragging force (Equation 2) acting on the liquid resin receding contact line, where extra curing due to residual can be partly suppressed comparing with vat polymerization” in Line 9, Page 15 of the revised manuscript.

4. The experimental setup not described in the manuscript.

Reply: Thanks for the reviewer’s suggestion. The experimental setup is a self-made bottom-up DLP printing. Accordingly, we have revised the text “The self-built 3D printing setup involves a UV projector, a UV-transparent curing interface, and an aluminum supporting plate mounting on a programmable moving platform from bottom to up based on our previous investigations^{25,26}” in Line 10, Page 4 in the revised manuscript. Accordingly, the detailed composition of the experimental setup has been added as the “**3D printing apparatus**” section in the **Methods** part of the revised manuscript.

5. Too much information is “hidden” in supplementary information and should be moved to the main manuscript if deemed essential.

Reply: Thanks for the reviewer’s suggestion. Considering the printing speed, precision, resin utilization efficiency are all essential factors for the 3D printing process, we have moved the data that “one-droplet 3D printing process can suppress excessive curing in the x-y plane which implies a better resolution” from the supplementary information to Figure 2 with corresponding descriptions on Page 7 - Page 8 in the revised manuscript, and Supplementary Figure 6 - 7 in the revised Supplementary Information.

6. The discussion on contact angles pp 7-8 becomes very confusing without a figure to support the descriptions.

Reply: Thanks for the reviewer’s suggestion. To help understanding the interfacial energies on Page 7-8, we have added a new scheme to describe the relationship among the three involved interfaces during the one-droplet 3D printing process (Figure R10). The new scheme is added as Figure 3b in the revised manuscript. Accordingly, the original Figure 2h is revised and moved to Figure 3a in the revised manuscript.

Figure R10 (Figure 3a-b in the revised manuscript). **a.** Scheme of the three interfaces involved in the one-droplet 3D printing process, including the interface between the liquid resin and the cured resin (1), the interface between the cured resin and the curing interface (2), and the interface between the liquid resin and the solid resin (3). **b.** Scheme of the two criteria of the one-droplet 3D printing process.

Captions of supplementary movie for Responses to Reviewers

Movie R1. Real-time monitoring of the curing of the “dark” region which connects to the patterned region.

Movie R2. Real-time monitoring of the repeated utilization of vat resin during the vat polymerization process.

Responses to Reviewer # 2

In this manuscript, the authors develop a novel strategy for improving the material utilization efficiency. The utilization efficiency of the materials is of importance for 3D printing. Currently, the common methods to improve the material efficiency mainly focus on optimizing the geometry structure or enhancing the printing processes. Here the authors provide an alternative method that is very interesting, simple and useful. The authors proposed an interesting concept by a one-droplet 3D printing process. Through focusing on the interfacial adhesion, the curing interfaces property and the droplet TCL shrinking process can be well controlled. As they mentioned, this method has achieved the record material utilization efficiency and high printing speed. This research is very useful for practical applications. Therefore, the reviewer would like to recommend the acceptance of this manuscript for publication in Nature Communications after minor revision suggested below.

Reply: The authors thank the reviewer very much for the positive comments and the suggestions for improving the manuscript. According to the reviewer's comments, we have made revisions on our manuscript and addressed them one by one as below.

1. Can UV resin with different surface tensions be versatile to this method?

Reply: Thanks for the reviewer's comment. In addition to the resin used in Figure 1-3 (Resin 1) and the resin used in Figure 4 (Resin 2), we further test another commercial resin (Resin 3 in the revised Supplementary Information, Flexible Resin, Formlabs, America) for the capability of the one-droplet 3D printing strategy, the process is displayed in Figure R11.

Figure R11 (Supplementary Figure 11 in the revised Supplementary Information). Optical captures of the one-droplet 3D printing process of a commercial flexible resin (Flexible Resin, Formlabs, America).

For the three tested resins, we have summarized the surface tension and the interfacial adhesions of the three involved interfaces, as displayed in Table R1. The one-droplet 3D printing tendency of the three resins is in consistence with the two criteria ($\gamma_1 > \gamma_3$ and $\gamma_3 > \gamma_2$ for three involved interfaces) that we summarized in the manuscript. For the versatility of this method, we think that if the involved interfacial adhesions can satisfy the summarized criteria, one-droplet 3D printing can be realized.

Table R1 (Supplementary Table 5 in the revised Supplementary Information). The summary of surface tensions and interfacial adhesions of different resin systems on the S-PDMS curing interface.

Resin	Surface Tension (mN/m)	Interfacial Adhesion (mJ/m ²)		
		Cured Resin /Curing Interface (γ_2)	Liquid Resin /Curing Interface (γ_3)	Liquid Resin /Cured Resin (γ_1)
Resin 1	31.1 ± 1.2	1.8 ± 0.3	56.9 ± 4.3	59.4 ± 2.9
Resin 2	35.8 ± 0.6	0.9 ± 0.5	54.9 ± 0.4	67.1 ± 0.8
Resin 3	32.1 ± 0.8	0.3 ± 0.2	51.8 ± 0.4	52.6 ± 0.7

Note: Resin 1, Resin 2, and Resin 3 are the resin used in Figure 1-3, the resin used in Figure 4, and the commercial flexible resin, respectively.

Accordingly, we have added the sentence “We further test the one-droplet 3D printing capability of a commercial flexible resin (Flexible Resin, Formlabs, America) on the S-PDMS surface (Supplementary Figure 11). The relationship of adhesions among the three involved interfaces is consistent with the two criteria mentioned above (Supplementary Table 2-5), which further proves our conclusion” in Line 20, Page 10 in the revised manuscript.

2. Would the lubricant leak from the curing interface to the printed structure? Then how to keep the lubricating liquid should be described in detailed.

Reply: Thanks for the reviewer’s suggestion. As ethanol is an excellent solvent to dissolve uncured resin during the post-processing for the cured structure. Here, we adapt the same way to test the reservation ability of the lubricant. Besides, ethanol washing of the pure PDMS and the silicone oil swelled PDMS substrates are conducted for comparison. Through immersing corresponding substrates into ethanol with magnetic stirring for 10 minutes as interval for each washing process, the substrates are taken out and weighed.

Figure R12. The weight ratio of PDMS, silicone oil swelled PDMS, and S-PDMS after violently washing with ethanol.

As displayed in Figure R12, the weight ratio (the ratio between the weight after several times' washing and the initial weight) of different samples is measured. Increasing the immersing times, the weight increases for PDMS and keeps almost the same for S-PDMS. The weight increase of PDMS is due to the swell of ethanol into the PDMS. While weight decreases with the increasing of washing times for the silicone oil swelled PDMS surface, which can be ascribed from the gradual washing away of silicone oil by ethanol. The S-PDMS remained the same weight, which means that ethanol can neither swell the S-PDMS surface nor takes the perfluoro-carbon away, indicating the superior lubricant reserving ability. As the perfluorocarbon used for S-PDMS surface is almost immiscible to any other liquid, it also endows S-PDMS surface with stability for 3D printing. In our investigation, the S-PDMS surface is stored with a thin layer of perfluorocarbon liquid covering it for storing. Accordingly, the storing method is added in the **Methods** section of the revised manuscript: "The lubricant-infused surface is stored by covering with a thin layer of lubricant".

3. The influence of contact angle and adhesive energy on the 3D printing is not clear, the authors should explain it in the manuscript.

Reply: Thanks for the reviewer's comment. We are sorry for missing the detailed procedures of the OWRK (Owens, Wendt, Rabel and Kaelble) method. Accordingly, we have added the parameters of the probe liquids, the measured contact angles of probe liquids on the tested surfaces, and the surface energies of the tested surfaces, which are concluded as Supplementary Table 1, Supplementary Table 2 and Supplementary Table 3, respectively. Based on the OWRK method, the interfacial adhesions between the liquid resin and the cured resin or curing interfaces are summarized in Supplementary Table 4 and Supplementary Table 5.

Comparing with the contact angle, the relative value of interfacial energies plays a more critical role in determining the one-droplet 3D printing method. Accordingly, we have added the text, "Based on the experiments demonstrated above, the interfacial adhesions are more critical for one-droplet 3D printing comparing with the curing interface's contact angle" in Line 1, Page 9 in the revised manuscript. Besides, we have added a new scheme (Figure 3b in the revised manuscript) of the three involved interfacial adhesions to more clearly demonstrate the influence of adhesive energy on the one-droplet 3D printing.

4. Is the candle soot-based hierarchical surface optically transparent? The optical characterization of the candle soot-based hierarchical surface should be supplemented.

Reply: Thanks for the reviewer's comment. The candle soot-based hierarchical surface is optically transparent, which will not influence the UV curing process. We have characterized the optical properties of the F-Quartz surface, the S-PDMS surface and the superamphiphobic surface through putting the three surfaces on a blue color image. As shown in Figure R13, the image is not influenced by the covering of the three substrates, which displays the optical transparency of the three tested surfaces. Accordingly, the optical characterization results have been added as Supplementary Figure 4 in the revised Supplementary Information with the description of "All three substrates have high

UV transparency, as displayed in Supplementary Figure 4” in Line 13, Page 5 in the revised manuscript.

Figure R13 (Supplementary Figure 4 in the revised Supplementary Information). Optical characterization of different curing interfaces. **a**, **b** and **c** are optical images of the superamphiphobic surface, the S-PDMS surface and the F-Quartz surface, respectively. **d**. Optical image of the three surfaces with water droplets sitting on each surface.

5. The reviewer is a little bit confused about whether some liquid resin is left on the supporting plate? As the supporting plate should be high adhesive to ensure the printing of the first layer.

Reply: Thanks for the reviewer’s comment. Some liquid resin is indeed left on the supporting plate, as shown in Figure 1b₃, which leads to the wet material utilization efficiency is not complete 100% (Figure 3c-f in the revised manuscript). As the supporting plate for DLP 3D printing is always selected from metals for their high surface energies, an aluminum plate is used in this work. The residual left on the supporting plate is quite little, for example, for printing Figure 1b, a 24-mm-long cured grid structure with a cylindrical shape, the resin utilization efficiency is 99.6%. Only 0.4% liquid resin is left on the supporting plate.

Accordingly, we have added the text “The remaining 0.4% of liquid resin is left on the supporting plate due to the adhesion between liquid resin and supporting plate (as displayed in Figure 1b₃)” in Line 6, Page 4 in the revised manuscript.

6. Several abbreviations need to be spelled out before using them (For example, OWRK on P. 7, UV on P. 1)

Reply: Thanks for the reviewer’s comment. The full name of the OWRK method, Owens, Wendt, Rabel and Kaelble method (Line 10, Page 9), and the full name of UV, ultraviolet (Line 15, Page 1), have been added in the revised manuscript.

Responses to Reviewer # 3

The paper presents an interesting approach to a Vat Polymerization process. The main claims are related to material utilization improvement and cost reduction. However, it can be expected that these aspects are less important for products of the size associated with this process type. It is also stated that an improved level of detail can be expected, which might be of greater importance for small (medical) artifacts. It would strengthen the paper if the claims in the direction of improvements for the end-user (especially costs and accuracy improvement) for a typical product (for example the tooth) are supported by numbers. Detailed comments, supporting the above, are included in the pdf.

Reply: Authors thank the reviewer very much for the interest in our results, and thank you for the suggestions to improve and strengthen our work. According to the reviewer's comments, we have made revisions on our manuscript and addressed them one by one as below. The suggestions indeed improve the quality of the revised manuscript. Thanks again for the reviewer's comments and suggestions.

1. Please emphasize how regulatable is executed.

12 **Abstract**

13 We propose a one-droplet 3D printing strategy to fabricate controllable 3D structures from a single
14 droplet with regulatable material utilization efficiency. The three-phase contact line of the resin droplet
15 recedes during the transformation of a droplet to a
16 desired 3D structure. The properties of the UV curing interface, which reduces

Reply: Thanks for the reviewer's comment. There are two material utilization efficiencies, including the wet material utilization efficiency (the weight ratio of the just printed structure to the initial liquid resin) and the net material utilization efficiency (the weight ratio of the acquired dry structure to the initial liquid resin). We are sorry for the omission of the wet material utilization efficiencies in Figure 3. Accordingly, we have added the data of wet material utilization efficiencies in Figure 3c - 3f in the revised manuscript, which indicates that the wet material utilization efficiency is larger than 96% for all tested processes. The net material utilization efficiency is regulatable, and the degree of the regulation is related to the contact area, the 3D distribution and the dewetting force of liquid resin on the cured structure, which depends on the printing and dewetting process as shown in Figure 3c - 3f in the revised manuscript. For example, as displayed in Figure 3c in the revised manuscript, with resin weight increasing from 24.3 mg to 53.4 mg, 75.2 mg, 100.4 mg and 150.9 mg, the net material utilization efficiency decreases from 94.8% to 94.5% to 93.7% and 90.1%, respectively.

2. In the main text this is mentioned as nearly 100%.

15 recedes during the UV curing process, which leads to the complete transformation of a droplet to a
16 desired 3D structure. The key factor is the slippery properties which reduces
17 the adhesion properties among the liquid resin, cured resin regulating the

Reply: Thanks for the reviewer’s comment. We are sorry for the expression of the value of the wet material utilization efficiency. Accordingly, the descriptions about the wet material utilization efficiency have been revised to the specific experimental value of the wet material utilization efficiency in the revised manuscript.

The changes are displayed as below:

(1). In Line 6 Page 11 of the revised manuscript, the sentence “Taking S-PDMS as an example, the wet material utilization efficiency (the weight ratio of the just cured structure before cleaning to the initial liquid resin) is nearly 100% for all tested processes” has been revised to “Taking S-PDMS as an example, as shown in Figure 3c-3f, all the wet material utilization efficiencies are above 96% for all tested structures ascribing to the receding TCL”.

(2). In Line 7, Page 15 of the revised manuscript, the sentence “The wet material utilization efficiency (87.5%) and dry material utilization efficiency (66.7%) are both much lower than those obtained with the one-droplet 3D printing process presented in this work (~100% for wet and 92.5% for dry material efficiency, Figure 3d, the length-to-width ratio of 17:1, 30°)” has been changed to “Both the wet material utilization efficiency (~87.5%) and the net material utilization efficiency (~66.7%) are much lower than those from the one-droplet 3D printing process (~98.6% for wet and 92.5% for net material efficiency, Figure 3f, L/W ratio of 17:1, 30°)”.

Besides, this sentence has been revised to “We propose a one-droplet 3D printing strategy to fabricate controllable 3D structures from a single droplet ascribing to the receding property of the three-phase contact line (TCL) of the resin droplet during the ultraviolet (UV) curing process” in Line 13, Page 1.

3. Could you indicate what these new avenues are? Also from the main text the strategic advantages should be better emphasized.

20 residue and high material utilization efficiency in forming a droplet into a 3D structure. This strategy
21 to construct 3D structures from a single droplet will open new avenues for controllable 3D
22 manufacturing.

Reply: Thanks for the reviewer’s question and suggestion. The key of this strategy is that the receding TCL endows free contact surfaces and low adhesions among involved surfaces, which leads to high material utilization efficiency and improved printing precision. In addition to the potential application of one demand teeth structure fabrication stated in the manuscript, one-droplet 3D printing can also be applied to fabricate eyeglass structure, even the contact lens structure (the figure displayed below), if further investigations are conducted. Also, we want to say that the application scope of one-droplet 3D printing is not limited by the introduction of the receding TCL, and can be applied to what vat polymerization can be applied to, but with higher material utilization efficiency and higher printing precision.

[redacted]

In the revised manuscript, we have conducted new experiments involving the comparison with vat polymerization to distinguish the priority of the one-droplet 3D printing strategy. As a prerequisite, we want to emphasize that continuous 3D printing is different from intermittent 3D printing process (including intermittent vat polymerization and intermittent one-droplet printing processes). During intermittent 3D printing, UV is on when curing, and is off during the interval. Traditionally, the interval is employed for cured resin separation, heat dissipation of the liquid resin, and the resin refilling between the cured resin and the curing interface for the generation of the next curing layer. While during continuous 3D printing, UV is always on with continuous projection of UV patterns onto the curing interface during the whole process, which leads to the continuous irradiation-induced accumulation of energy at the curing interface inside the liquid resin. In addition to the requirement of sufficient heat dissipation, the requirement of resin mobility, which influences the generation of a new curing interface, is also more critical for continuous printing than intermittent printing. All experiments are conducted without damaging the curing interface, *i.e.*, the results are concluded with the exclusion of the influence of the curing interface.

Figure R1 (Supplementary Figure 1 in the revised Supplementary Information). Afterglow of the “dark” region of the UV projector under different UV intensities. **a.** Scheme of the patterned region and the non-patterned region of the UV pattern projected from the UV source. The star shape is the tested pattern and the white frame represents the breath of the UV projector. **b - i.** Optical images of the UV pattern when projecting the pattern in **a** with different UV intensities (from 1 mW/cm² to 40 mW/cm²). The images are captured with the camera settings (including the exposure time, magnification, photosensitivity and aperture, *et al.*) unchanged. The UV projector can provide 405 nm pattern with the projection area of 51.6 mm × 32.2 mm. With the increasing of the light intensity, the afterglow across the whole non-patterned dark region increases significantly, which finally may lead to the curing of the non-patterned dark region.

First, the afterglow of the “dark” region of the UV pattern will pollute or even cured the vat resin. Traditionally, a slice of UV pattern contains the patterned region and the non-patterned region (Figure R1a), where the UV patterned region displayed light of predetermined intensity, and the non-patterned region should be completely dark or 0 mW/cm². However, current DLP light source is based on LED. The “dark” region, *i.e.*, the non-patterned region of a projected slice, is not really 100% “black” or 0 mW/cm² UV intensity. As shown in Figure R1b - 1i, with the increasing of the UV intensity, the afterglow of the “dark” region becomes brighter and brighter. Just as we mentioned above, the continuous printing process requires the continuous projection of the UV light with a certain intensity. The afterglow of the “dark” region will lead to the unintentional generation of free radicals and result in the inhomogeneous composition of the vat resin. The amount of free radical is not generated quantitatively, and the worst condition is the trigger of the curing of the un-patterned “dark” region.

Figure R2 (Supplementary Figure 2 in the revised manuscript). The unintentional curing of the “dark” region during the vat polymerization process. **a.** Optical image of the vat after continuous printing the typical pattern (1 mm round shape) characterized in the Figure R7 (UV intensity of 30 mW/cm^2 and printing speed of $100 \text{ }\mu\text{m/s}$). The blue dot and the square black dotted frame are scheme of the UV pattern and the breadth of the UV projector, respectively. **b.** Optical image of the unintentionally cured resin. **c-e.** Series of optical captures of the curing of the “dark” region which influences the printing precision. Insets are enlarged view of corresponding regions. Red circles indicate the cured part of the “dark” region which connects to the patterned region.

As shown in Figure R2, after continuous printing the typical pattern (1 mm round shape with UV intensity of 30 mW/cm^2 and printing speed of $100 \text{ }\mu\text{m/s}$) used in the manuscript, the non-patterned region is unintentionally cured, which cannot be easily recognized without leaning the vat. Besides, for printing 3D structures with a large area of UV patterned regions, the cured part by the afterglow may be connected with the UV patterned region (Figure R2c-e, Movie R1 which real-time monitors the curing of the “dark” region which connects to the patterned region), which will further influence the printing precision. Therefore, for continuous 3D printing process, the continuous projection of the UV light will pollute the resin non-quantitatively and will affect the printing precision.

Figure R3 (Supplementary Figure 3 in the revised Supplementary Information, Movie R2). Optical captures of the cured structure after repeatedly using the same vat resin through vat polymerization. The experiment is

conducted through repeating utilizing the same vat liquid resin at the same position with the same UV pattern, UV intensity and printing speed (50 mW/cm^2 , $200 \text{ }\mu\text{m/s}$).

Second, we have found that the repeating use of the vat resin will decrease the 3D printing precision. We have repeatedly used the same location on the curing interface to continuously print the same structure from the same vat resin for five times (50 mW/cm^2 , $200 \text{ }\mu\text{m/s}$). For the first time, the newly added liquid resin can print a relatively uniform structure. With the increase of the repeating time, the uniformity of the printed structure decreased, which leads to rough protruding on the structure sidewall (yellow circles in Figure R3, Movie R2 which real-time monitors the repeated utilization of vat resin during the vat polymerization process). Therefore, our proposed one-droplet 3D printing strategy not only avoids the reutilization of vat resin but also can solve this problem intrinsically.

Figure R4 (Supplementary Figure 9 in the revised Supplementary Information). The unavailability of vat resin for curing even with a few printing layers high. **a-e.** Optical captures of the vat polymerization process of the gyroid structure with the dimension in the x-y plane of $4.5 \text{ cm} \times 2.3 \text{ cm}$ (5 mW/cm^2 , printing speed of $20 \text{ }\mu\text{m/s}$). The red lines in **a-d** are outlines of the upper surface of the vat resin.

Third, the vat resin is confined by the vat sidewall and the vat bottom, which leads to the confined liquid mobility and the unavailability of liquid resin as resin source for curing even with a few printing layers high and under a low printing speed. We have compared the liquid resin mobility of the vat polymerization with one-droplet printing process under the condition when the amount of vat resin is theoretically available to print the designed 3D structure. As displayed in Figure R4, the TCL of the vat resin is pinned on the sidewall and the bottom of the vat. With the

continuous consumption of the vat resin, the upper surface of the liquid resin distorted with the upper surface of the liquid resin near the sidewall higher than the inner surface, as displayed by the red dotted lines in Figure R4a - d. Finally, the resin near the cured structure is completely consumed, while the resin near the vat sidewall cannot be utilized, which leads to the failure of further printing. Therefore, the liquid remains inside the vat with certain resin thickness because of the confinement of the contact surface by the vat, so that the vat resin cannot be utilized even with a few printing layers high.

Figure R5 (Supplementary Figure 10 in the revised Supplementary Information). One-droplet 3D printing a gyroid structure. **a-e.** Optical captures of the one-droplet printing process of the gyroid structure with the dimension in the x-y plane of 4.5 cm × 2.3 cm. **f.** Optical capture of the printed structure after UV off. **g.** Optical image of the supporting plate employed for printing the gyroid structure. **h.** Optical image of the gyroid structure 3D printed from one-droplet.

While for one-droplet 3D printing of the same structure, as displayed in Figure R5, the TCL of the sizeable liquid resin droplet can continuously recede along with the continuous printing process, which proves the freer contact surfaces or higher TCL mobility of the one-droplet printing process. Therefore, the resin mobility, *i.e.*, the regeneration speed of a new curing interface, is high for one-droplet 3D printing process to maintain a continuous printing process.

Figure R6 (Supplementary Figure 8 in the revised Supplementary Information). Optical tracking of the inner liquid flow during the continuous one-droplet 3D printing process through adding lightweight and black carbon nanofibers inside the liquid resin droplet. **a.** The initial position of Particle A (red circle) in the liquid resin droplet. **b.** The initial position of Particle B and part of the trajectory of particle A. Blue circle represent the initial position of Particle B. **c.** The trajectories of particle A and particle B in the liquid resin and cured resin structure. Red and blue dot lines indicate the trajectories of particle A and particle B, respectively. Red dot lines indicate the trajectories of particle A and particle B, respectively. **d.** The trajectories of particle A and particle B in the liquid resin and cured resin structure. Red and blue dot lines indicate the trajectories of particle A and particle B, respectively.

Fourth, the liquid motion inside the droplet and at the curing interface is vigorous during the continuous printing process. We have added immiscible additive inside the transparent resin droplet to track the internal liquid flow inside the droplet during the continuous printing process. Carbon nanofibers are selected as the additive, as their color is black for easy tracking, and their weight is light enough to flow along with the internal liquid flow easily. As shown in Figure R6 (Supplementary Movie 3), with the continuous lifting of the supporting plate, the inner droplet liquid can flow along the liquid resin-air interface upward to the cured structure-liquid resin interface (the trajectory of Particle A in Figure R6a-c) driven by the adhesion of γ_1 , or along the sidewall of the cured structure downward to the curing interface (the trajectory of Particle A in Figure R6c-d and the trajectory of Particle B in Figure R6c) driven by the adhesion of γ_3 . The liquid resin inside the whole droplet is thus continuously moving, and the liquid resin is mixed uniformly during the continuous printing process.

Figure R7 (Supplementary Figure 6a-b in the revised Supplementary Information and Figure 2h-i in the revised manuscript). Comparison of the continuous printing process of the vat polymerization and the one-droplet 3D printing process (printing speed of 100 $\mu\text{m/s}$, 500 s duration for printing structure of 5 cm high). **a.** Series of optical captures of the one-droplet 3D printing process. **b.** Series of optical captures of the vat polymerization process. **c.** The time-variant width change of the left and right edges on the structure constructed from one-droplet continuous printing, with the vat polymerization process as control. Red and black lines represent the width change curves of the one-droplet printing process and the vat polymerization process, respectively. Circles and squares represent the left and right edges of the printed structure, respectively. **d.** The time-variant D/D_{design} ratio change of the one-droplet printing process, with the vat polymerization process as control. Red and black lines represent D/D_{design} ratio change curves of the one-droplet printing process and the vat polymerization process, respectively. Scale bars are 1 mm.

Fifth, extra curing during continuous printing process can be prevented through one-droplet printing. We directly monitor the *in situ* curing process of vat polymerization and one-droplet printing process in real-time with the same experimental conditions to quantitatively compare the printing precision and evidence the improved precision. Two parameters, including the printing width of the left and right edges which reflects the outer profile of the printed structure, and the ratio (D/D_{design}) of the printed width (D) to the designed width (D_{design}) which reflects the extra curing phenomenon, are characterized for comparison. As displayed in Figure R7a and the red dotted lines in Figure R7c, the printing process is stable with the left and right edges keeping straight during the whole process (left edges range from $-492.9 \mu\text{m}$ to $-507.1 \mu\text{m}$, right edges range from $492.9 \mu\text{m}$ to $500.0 \mu\text{m}$), as well as the stable D/D_{design} ratio (remained between 0.986 and 1.007, the red dotted line in Figure R7d). While for vat polymerization, the printing width becomes nonuniform (left edges range from $-259.1 \mu\text{m}$ to $-663.4 \mu\text{m}$, right edges range from $445.4 \mu\text{m}$ to $704.5 \mu\text{m}$) with protruding or stepped structures on the sidewall along with the increasing of printing time (Figure R7b and the black dotted lines in Figure R7c). The D/D_{design} ratio also becomes larger (from 0.981 to the maximum of 1.336 during the 500 s continuous curing process) and unstable with the increasing of the time (black dotted line in Figure R7d). The real-time characterization of the two processes is displayed in Supplementary Movie 2.

Figure R8 (Supplementary Figure 7 in the revised Supplementary Information). Comparison of the contact surface mobility of the one-droplet 3D printing and the vat polymerization process. **a.** Scheme of the configuration of the one-droplet 3D printing process. **b.** Scheme of the free contact surfaces during the one-droplet 3D printing process. Blue lines are the outlines of the free contact surfaces. **c.** Scheme of the configuration of the vat polymerization process. **d.** Scheme of the confined contact surfaces of the vat polymerization process. Purple lines are the confined contact surfaces.

The difference in the printing process and the printing precision can be due to the difference in the confinement effect of the contact surfaces. For the one-droplet 3D printing process, the liquid resin can recede during the curing process (Figure R8a-b). The contact surface between the liquid resin and the curing interface, and the contact surface between the liquid resin and the air are free. It endows a sufficient resin mobility on the curing interface for continuous UV curing. Therefore, the one-droplet printing process is dominated by the relationship among the three interfaces, as described in Figure 3a-b in the revised manuscript. The sufficient inner liquid mobility can also be supported by the vigorous inner liquid flow inside the liquid resin droplet (Figure R6), which leads to a stable and continuous one-droplet printing process.

In contrast, for vat polymerization, the contact surface between the liquid resin and the curing interface, and the contact surface between the liquid resin and the air are confined by the vat. Under this confinement, the vat resin prefers to be cured rather than be freely moving to endow a sufficient curing interface for UV curing (Figure R8c-d). Extra curing with protruding or stepped structures on the sidewall or with bubbles generating inside the cured resin occurs during the continuous printing process (Figure R7b), which results in printing instability and decrease in printing precision (black dotted line in Figure R7d). Besides, the confinement of the resin by the vat can also be reflected in the unavailability of the vat resin to serve as resin source for curing even with a few printing layers high, but with the complete consumption of the resin near the cured structure while higher resin upper surface near the vat sidewall (Figure R4). And the availability of one-droplet strategy to print the same structure (Figure R5). Therefore, the confinement of the liquid resin to a droplet morphology with receding TCL can suppress excessive curing in the x-y plane, and improve the printing accuracy and stability of the continuous 3D printing process.

Figure R9 (Supplementary Figure 18 in the revised Supplementary Information). Micro-CT characterization of the V-grooved structures prepared from vat polymerization and one-droplet 3D printing. **a**, **b** and **c** are inner side view of the designed, Micro-CT image of V-grooved structure prepared from vat polymerization and Micro-CT image of V-grooved structure prepared from one-droplet 3D printing, respectively. **d**, **e** and **f** are outer side view of the designed, Micro-CT image of V-grooved structure prepared from vat polymerization and Micro-CT image of V-grooved structure prepared from one-droplet 3D printing, respectively. **g**, **h** and **i** are cross-sectional view of the designed, Micro-CT image of V-grooved structure prepared from vat polymerization and Micro-CT image of V-grooved structure prepared from one-droplet 3D printing, respectively. The blue V patterns in **h** and **i** are designed V patterns. Scale bars are 2 mm.

Sixth, sharper inner V-corner can be printed through one-droplet printing than vat polymerization. We have compared the detailed morphologies of the asymmetric V-groove structures (with intersection angle of 30°) printed from one-droplet 3D printing (wet material utilization of $\sim 98.6\%$ and net material utilization efficiency of $\sim 92.5\%$, Figure 3f in the revised manuscript) and from vat polymerization (wet material utilization of $\sim 87.5\%$ and net material utilization efficiency of $\sim 66.7\%$, Supplementary Figure 13) through Micro-CT characterization. As shown in Figure R9, one-droplet 3D printing process can print sharper inner V-corner than vat polymerization. This can be due to the dragging force (Equation 2 in the manuscript) acting on the liquid resin contact line, which leads to higher net material utilization efficiency and less residual

inside the V-corner. In other words, extra curing due to residual at the inner V-corner can be partly suppressed during one-droplet 3D printing process. Therefore, one-droplet 3D printing process with higher material utilization efficiencies not only can reduce the residual, but also can increase the printing precision.

Figure R11 (Supplementary Figure 11 in the revised Supplementary Information). Optical captures of the one-droplet 3D printing process of the commercial flexible resin (Flexible Resin, Formlabs, America).

Seventh, one-droplet 3D printing method is methodologically versatile to resins of different surface tensions. In addition to the resin used in Figure 1-3 (Resin 1) and the resin used in Figure 4 (Resin 2), we further test another commercial resin (Resin 3 in the revised Supplementary Information, Flexible Resin, Formlabs, America) for the capability of the one-droplet 3D printing strategy, the process is displayed in Figure R11. For the three tested resins, we have summarized the surface tension and the interfacial adhesions of the three involved interfaces, as displayed in Table R1. The one-droplet 3D printing tendency of the three resins is in consistence with the two criteria ($\gamma_1 > \gamma_3$ and $\gamma_3 > \gamma_2$ for three involved interfaces) that we summarized in the manuscript. For the versatility of this method, we think that if the involved interfacial adhesions can satisfy the summarized criteria, one-droplet 3D printing can be realized.

Table R1 (Supplementary Table 5 in the revised Supplementary Information). The summary of surface tensions and interfacial adhesions of different resin systems on the S-PDMS curing interface.

Resin	Surface Tension (mN/m)	Interfacial Adhesion (mJ/m ²)		
		Cured Resin /Curing Interface (γ_2)	Liquid Resin /Curing Interface (γ_3)	Liquid Resin /Cured Resin (γ_1)
Resin 1	31.1 ± 1.2	1.8 ± 0.3	56.9 ± 4.3	59.4 ± 2.9
Resin 2	35.8 ± 0.6	0.9 ± 0.5	54.9 ± 0.4	67.1 ± 0.8
Resin 3	32.1 ± 0.8	0.3 ± 0.2	51.8 ± 0.4	52.6 ± 0.7

Note: Resin 1, Resin 2, and Resin 3 are the resin used in Figure 1-3, the resin used in Figure 4, and the commercial flexible resin, respectively.

In conclusion, as the employed liquid resin droplet is cured to a designed 3D structure, the one-droplet 3D printing method avoids the repeatable irradiation of liquid resin with UV light, neither by the UV pattern nor by the afterglow of the “dark” region. Moreover, the receding TCL endows higher liquid mobility both inside the droplet and at the curing interface during the

continuous printing process. This leads to the uniform mixing of the liquid resin, inhibits extra curing and results in stable curing during the continuous printing process with straight outlines and stable D/D_{design} ratio. With higher material utilization efficiencies, sharper inner V-corner of V-grooved structure can be printed than vat polymerization, which further proves the increased printing precision and the strategic advantage of one-droplet 3D printing. Corresponding experimental results have been added in the revised manuscript and the revised Supplementary Information as figures and corresponding descriptions.

4. Please define material utilization efficiency. The definitions are presented in the text, but way after they are used.

Reply: Thanks for the reviewer’s comment. We have revised this sentence through adding the definition of corresponding material utilization efficiency to “Although effective in constructing fine structures, compared with fused deposition modeling (FDM), the existing digital light processing (DLP), stereolithography (SLA) printing and volumetric additive manufacture technologies have low wet material utilization efficiency (the weight ratio of the just printed structure to the initial liquid resin) and net material utilization efficiency (the weight ratio of the acquired dry structure to the initial liquid resin)” in Line 6, Page 2. In addition, we have revised the expression of material utilization efficiency throughout the manuscript and indicated whether it is wet or net when it appears.

5. For bottom up vat polymerization techniques this is not the case. The layer of resin has to be just a few print layers high. So large is very relative. For the sizes associated with products suitable for one-drop printing this comes down to 10 to 100 of cubic milliliters of resin.

Reply: Thanks for the reviewer’s comment. As displayed in Supplementary Figure 13 in the revised Supplementary Information, the vat polymerization we compared is the bottom-up configuration, both the wet and net material utilization efficiencies are low, meaning that this problem also occurs in the bottom-up vat polymerization techniques. As displayed in Figure R4, the TCL of the vat resin is pinned on the sidewall and the bottom of the vat. With the continuous consumption of the vat resin,

the upper surface of the liquid resin distorted with the upper surface of the liquid resin near the sidewall higher than the inner surface, as displayed by the red dotted lines in Figure R4a - d. Finally, the resin near the cured structure is completely consumed, while the resin near the vat sidewall cannot be utilized, which leads to the failure of further printing. Therefore, the liquid remains on the vat with certain resin thickness because of the confinement of the contact surface by the vat, so that the vat resin cannot be utilized even with a few printing layers high and under a low printing speed.

Figure R4 (Supplementary Figure 9 in the revised Supplementary Information). The unavailability of vat resin for curing even with a few printing layers high. **a-e.** Optical captures of the vat polymerization process of the gyroid structure with the dimension in the x-y plane of $4.5\text{ cm} \times 2.3\text{ cm}$ ($5\text{mW}/\text{cm}^2$, printing speed of $20\text{ }\mu\text{m}/\text{s}$). The red lines in **a-d** are outlines of the upper surface of the vat resin.

According to the reviewer’s comment, we have revised the text from “First, a large amount of uncured resin needs to fill the tank before the printing process” to “Uncured resin needs to cover the whole tank with an excessive quantity before the printing process, which not only increases the resin cost but also leads to resin waste” in Line 11, Page 2 in the revised manuscript. Figure R4 is added in the revised Supplementary Information as Supplementary Figure 9 with the description of “Besides, the confinement of the resin vat can be reflected in the unavailability of the vat resin for curing even with a few printing layers high, *i.e.*, the resin near the cured structure can be completely consumed while the resin near the vat sidewall cannot be used (Supplementary Figure 9)” in Line 13, Page 8 in the revised manuscript.

6. You discuss the printing of small structures. For those, resin cost is minimal. The material that does not end up in the product is no waste, but can be reused in the next product.

9 technologies have low material utilization efficiency. First, a large amount of uncured resin needs to
 10 fill the tank before the printing process, which not only increases the resin cost but also leads to resin
 11 waste. Second, the residual resin on the surface of the substrate leads to a low printing resolution
 12 during the following printing process.

redacted 5月6日 回复 x
 You discuss the printing of small structures. For those, resin cost is minimal. The material that does not end up in the product is no waste, but can be reused in the next product.

Reply: Thanks for the reviewer’s comment. To prove that one-droplet 3D printing can also be utilized to print large 3D structures, we have printed a gyroid structure with the dimension in the x-y plane of 4.5 cm × 2.3 cm, which is relatively large for our UV projector with the breadth of 5.16 cm × 3.22 cm. As displayed in Figure R5, the TCL of the liquid resin droplet can continuously recede along with the continuous printing process. While for vat polymerization, the liquid resin cannot be utilized during the vat polymerization, even with a few layers of liquid resin inside the vat as displayed in Figure R4. Moreover, we have displayed that the unavoidable afterglow of “dark” region of UV projectors (Figure R1, Figure R2), and the repeated using of vat resin during vat polymerization (Figure R3) will lead to printing instability and decreased printing precision. Therefore, the one droplet 3D printing strategy can avoid repeated using of the vat liquid resin membrane, which avoids the widening and roughening of printed structure due to continuous and repeating irradiation.

Figure R5 (Supplementary Figure 10 in the revised Supplementary Information). One-droplet 3D printing a gyroid structure. **a-e.** Optical captures of the one-droplet printing process of the gyroid structure with the dimension in the x-y plane of 4.5 cm × 2.3 cm. **f.** Optical capture of the printed structure after UV off. **g.** Optical image of the supporting plate employed for printing the gyroid structure. **h.** Optical image of the gyroid structure 3D printed from one-droplet.

Accordingly, we have added corresponding experimental results as Supplementary Figure 10 in the revised Supplementary Information with corresponding description of “For printing the same structure through one-droplet 3D printing, TCL can smoothly recede with continuous curing, which

leads to the curing of the liquid droplet to a gyroid structure (Supplementary Figure 10)” in Line 16, Page 8 in the revised manuscript.

7. The resolution could be important for potential users. Could you add figures/measurements supporting the improved resolution.

11 waste. Second, the residual resin on the cured structure surface will decrease the 3D printing resolution
 12 during the following printing process, as the printed part is
 13 irradiation, where heat is accumulated during the one-droplet

Reply: Thanks for the reviewer’s comment. According to the reviewer’s suggestion, we directly monitor the *in situ* curing process of vat polymerization and one-droplet printing process in real-time with the same experimental conditions to quantitatively compare the printing precision and evidence the improved precision. Two parameters, including the printing width of the left and right edges which reflects the outer profile of the printed structure, and the ratio (D/D_{design}) of the printed width (D) to the designed width (D_{design}) which reflects the extra curing phenomenon, are characterized for comparison.

Figure R7 (Supplementary Figure 6a-b in the revised Supplementary Information and Figure 2h-i in the revised Manuscript). Comparison of the continuous printing process of the vat polymerization and the one-droplet 3D printing process (printing speed of 100 $\mu\text{m/s}$, 500s duration for printing structure of 5 cm high). **a.** Series of optical captures of the one-droplet 3D printing process. **b.** Series of optical captures of the vat polymerization process. **c.** The time-variant width change of the left and right edges on the structure constructed from one-droplet

continuous printing, with the vat polymerization process as control. Red and black lines represent the width change curves of the one-droplet printing process and the vat polymerization process, respectively. Circles and squares represent the left and right edges of the printed structure, respectively. **d.** The time-variant D/D_{design} ratio change of the one-droplet printing process, with the vat polymerization process as control. Red and black lines represent D/D_{design} ratio change curves of the one-droplet printing process and the vat polymerization process, respectively. Scale bars are 1 mm.

As displayed in Figure R7a and the red dotted lines in Figure R7c, the printing process is stable with the left and right edges keeping straight during the whole process (left edges range from $-492.9 \mu\text{m}$ to $-507.1 \mu\text{m}$, right edges range from $492.9 \mu\text{m}$ to $500.0 \mu\text{m}$), as well as the stable D/D_{design} ratio (remained between 0.986 and 1.007, the red dotted line in Figure R7d). While for vat polymerization, the printing width becomes nonuniform (left edges range from $-259.1 \mu\text{m}$ to $-663.4 \mu\text{m}$, right edges range from $445.4 \mu\text{m}$ to $704.5 \mu\text{m}$) with protruding or stepped structures on the sidewall along with the increasing of printing time (Figure R7b and the black dotted lines in Figure R7c). The D/D_{design} ratio also becomes larger (from 0.981 to the maximum of 1.336 during the 500 s continuous curing process) and unstable with the increasing of the time (black dotted line in Figure R7d). The real-time characterization of the two processes is displayed in Supplementary Movie 2.

Figure R8 (Supplementary Figure 7 in the revised Supplementary Information). Comparison of the contact surface mobility of the one-droplet 3D printing and the vat polymerization process. **a.** Scheme of the configuration of the one-droplet 3D printing process. **b.** Scheme of the free contact surfaces during the one-droplet 3D printing process. Blue lines are the outlines of the free contact surfaces. **c.** Scheme of the configuration of the vat polymerization process. **d.** Scheme of the confined contact surfaces of the vat polymerization process. Purple lines are the confined contact surfaces.

The difference in the printing process and the printing precision can be due to the difference in the confinement effect of the contact surfaces. For the one-droplet 3D printing process, the liquid resin can recede during the curing process (Figure R8a-b). The contact surface between the liquid resin and the curing interface, and the contact surface between the liquid resin and the air are free. It

endows a sufficient resin mobility on the curing interface for continuous UV curing. Therefore, the one-droplet printing process is dominated by the relationship among the three interfaces, as described in Figure 3a-b in the revised manuscript. The sufficient inner liquid mobility can also be supported by the vigorous inner liquid flow inside the liquid resin droplet (Figure R6), which leads to a stable and continuous one-droplet printing process.

In contrast, for vat polymerization, the contact surface between the liquid resin and the curing interface, and the contact surface between the liquid resin and the air are confined by the vat. Under this confinement, the vat resin prefers to be cured rather than be freely moving to endow a sufficient curing interface for UV curing (Figure R8c-d). Extra curing with protruding or stepped structures on the sidewall or with bubbles generating inside the cured resin occurs during the continuous printing process (Figure R7b), which results in printing instability and decrease in printing precision (black dotted line in Figure R7d). Besides, the confinement of the resin by the vat can also be reflected in the unavailability of the vat resin to serve as resin source for curing even with a few printing layers high, but with the complete consumption of the resin near the cured structure while higher resin upper surface near the vat sidewall (Figure R4). And the availability of one-droplet strategy to print the same structure (Figure R5). Therefore, the confinement of the liquid resin to a droplet morphology with receding TCL can suppress excessive curing in the x-y plane, and improve the printing accuracy and stability of the continuous 3D printing process.

Accordingly, we have added corresponding experimental results as Figure 2h-i in the manuscript, Supplementary Figure 6-7 in the revised Supplementary Information and Supplementary Movie 2 to evidence the better resolution of one-droplet 3D printing convincingly.

8. This sentence is unclear to me.

Reply: Thanks for the reviewer’s comment. We are sorry for the mistake of this sentence, and we have revised it to “Also, as the UV resin is exothermic, heat dissipation is not sufficient for continuous printing process²⁰, especially for high-speed printing that requires high UV intensity. Accompanying with the residual resin on the cured structure surface, and the continuous irradiation of the resin under exciting light including the afterglow of the UV projector, extra curing or printing instability occurs which will decrease the 3D printing resolution (Supplementary Figure 1 - Supplementary Figure 3)” in Line 13, Page 2 in the revised manuscript.

9. Please explain why there will be no residue on the substrate. What happens with the excess residue if the volume of the droplet was bigger than the volume of the intended structure.

16 structure (Step IV) without residue on the substrate. The continuous receding of the resin droplet TCL
 17 on the curing interface facilitates a printing process with a high resin utilization efficiency. As shown in
 18 Figure 1b, a 24-mm-long cured grid structure is printed with a resin utilization

Please explain why there will be no residue on the substrate. What happens with the excess residue if the volume of the droplet was bigger than the volume of the intended structure

Reply: Thanks for the reviewer’s comment. As the adhesion between the liquid resin and the curing interface is smaller than the adhesion between the liquid resin and the cured structure, the droplet TCL can recede on the curing interface along with the continuous curing process, which is the first criterion ($\gamma_1 > \gamma_3$) that should be satisfied for one-droplet 3D printing. Besides, $\gamma_1 > \gamma_3$ also means that liquid resin prefers to adhere to the cured structure rather than the curing interface, which leads to no residual on the substrate. If the volume of the droplet is bigger than the volume of the intended structure, a droplet of liquid resin, whose volume is smaller than the original one, will leave on the curing interface after printing, as shown in Figure R14.

Figure R14. Optical captures the one-droplet 3D printing process when the volume of the droplet is bigger than the volume of the intended structure.

10. Define this efficiency.

17 on the curing interface facilitates a printing process with a high resin utilization efficiency. As shown in
 18 Figure 1b, a 24-mm-long cured grid structure is printed with a resin utilization
 19 an efficiency of 99.6%.

Define this efficiency.

Reply: Thanks for the reviewer’s comment. Accordingly, we have revised this sentence to “As shown in Figure 1b, a 24-mm-long cured grid structure with a cylindrical shape is printed with a wet resin utilization efficiency of 99.6%” in Line 5, Page 4 in the revised manuscript.

11. Please add a reference to the source of this equation and its constants. Could the variables used also be included in fig 1 and/or 2?

8 retraction v_m scales as $v^* \theta_E^3 / 9\sqrt{3}l$, where v^* is the threshold capillary velocity and scales as μ/γ . l
 9 typically ranges from 15 to 20, and θ_E is the equilibrium contact angle. A larger contact angle means a
 10 larger upper limit of the threshold velocity. This can be adjusted by varying the chemical

Please add a reference to the source of this equation and its constants. Could the variables used also be included in fig 1 and/or 2?

Reply: Thanks for the reviewer’s comment. We are sorry for the missing of the reference of this equation. Accordingly, we have added reference 27 - 28 in the revised manuscript. The variables of v_m and θ_E are included in Figure 2a in the revised manuscript, as displayed in Figure R15.

Figure R15 (Figure 2a in the revised manuscript). Scheme of the experimental setup for the one-droplet 3D printing process. Inset is the scheme of the UV pattern employed in Figure 2.

12. This sentence is unclear to the reviewer. Maybe it could be reformulated.

12 sidewall are inevitably achieved on the superamphiphobic surface. The S-PDMS surface is thus the
 13 best choice for the one-droplet curing interface among the three typical low-adhesion surfaces studied,
 14 as cured resin adheres to the quartz-based surface, with rupture occurring at the supporting plate, and
 15 a vertical striped pattern is formed on the sidewall of the printed structure when using the
 16 superamphiphobic surface.

Reply: Thanks for the reviewer’s comment. We have revised this sentence to “In conclusion, comparing with the quartz-based surface with ruptured cured resin and superamphiphobic surface with vertical striped patterns formed on the cured resin sidewall, the S-PDMS surface is thus the best choice for one-droplet 3D printing” in Line 7, Page 7 in the revised manuscript.

13. Figure 2H shows an axisymmetric situation, so it could be better explained what all the surfaces are in 2D. Maybe include a 2D picture also.

19 generated at the corresponding interfaces are systematically analyzed. As shown in Figure 2h, three
 20 interfaces participate in the one-droplet 3D printing process
 21 and the cured resin, (2) the interface between the cured resin

Figure 2h shows an axisymmetric situations, so it could be better explained what all the surfaces are in 2D. Maybe include a 2D picture also

Reply: Thanks for the reviewer’s comment. According to the reviewer’s suggestion, we have added the light pattern used for printing in revised Figure 2a, as displayed in Figure R15. The corresponding description has been revised to “The light pattern is set as a round shape with a diameter of 1 mm (Figure 2a inset)” in Line 15, Page 4 in the revised manuscript.

Figure R15 (Figure 2a in the revised manuscript). Scheme of the one-droplet 3D printing apparatus. Inset is the scheme of the UV pattern employed in Figure 2.

14. Reference?

2 a constant value for a specific resin system ($59.4 \pm 2.9 \text{ mJ/m}^2$ in the utilized system). The adhesion
 3 between the cured resin and the curing interface (γ_2), *i.e.*, the work needed
 4 from the curing interface, is measured by immobilizing the supporting plate

Reference?

Reply: Thanks for the reviewer’s comment. In this manuscript, the adhesions between the liquid resin and the solid surfaces (including the cured resin and the curing interface) are acquired by the OWRK (Owens, Wendt, Rabel and Kaelble) method (Reference 38 - Reference 40 in the revised manuscript). We are sorry for the missing of the detailed process of the OWRK method and we have

added it in the revised Supplementary Information. In detail, the parameters of the probe liquids, the measured contact angles of probe liquids on the tested surfaces and the surface energies of the tested surfaces, have been concluded in Supplementary Table 1, Supplementary Table 2 and Supplementary Table 3, respectively. The interfacial adhesions between the liquid resin and the cured resin or curing interfaces are summarized in Supplementary Table 4 and Supplementary Table 5.

Corresponding descriptions have been revised to “The adhesion between the liquid resin and the cured resin (γ_1) determines the residual amount of liquid resin on the cured structure. It has a constant value for a specific resin system ($59.4 \pm 2.9 \text{ mJ/m}^2$ of the investigated resin), which can be acquired through the OWRK (Owens, Wendt, Rabel and Kaelble) method³⁸⁻⁴⁰. The detailed processes are illustrated in Supplementary Figure 20 and Supplementary Table 1-5” in Line 7, Page 9 in the revised manuscript.

15. measurement itself not described; Reference to the supplementary information?

9 process during the one-droplet printing process, can be calculated with the OWRK method³⁶⁻³⁸. The 
10 quartz ($69.4 \pm 1.4 \text{ mJ/m}^2$), F-quartz ($48.8 \pm 3.6 \text{ mJ/m}^2$) surfaces 
11 possess similar adhesion values, while the superamphiphilic surfaces possess a smaller value

12 ($3.7 \pm 0.6 \text{ mJ/m}^2$).

Reply: Thanks for the reviewer’s comment. Similar to the adhesion between the liquid resin and the cured resin, the adhesion between the liquid resin and the curing interfaces are also acquired by the OWRK method. Corresponding descriptions have been revised to “The adhesion between the liquid resin and the curing surface (γ_3), which determines the TCL receding process during the one-droplet printing process can also be acquired through the OWRK method” in Line 16, Page 9 in the revised manuscript.

16. These are values based on surface area’s. I understand it correctly i have two questions: Do these surface area’s play a role in this equation? Or in other words, are these surface area equal?

17 *i.e.*, $\gamma_1 > \gamma_3$, so that the TCL can keep receding  consumption

18 process, which determines whether the liquid  3D structure.

19 Experimentally, γ_1 is greater than γ_3 when the S-P-DIMS or superamphiphilic surface is used as the

Reply: Thanks for the reviewer’s comment. We also have recognized the matter of surface area that the reviewer concerned, where the interfacial energy between two surfaces (mJ) is surface area related. Therefore, to normalize the function of surface area and to compare the relative magnitude of the interfacial adhesions of the three involved interfaces, the energy per surface area (mJ/m^2), rather than the interfacial energy (mJ), is used to investigate the criteria of the one-droplet 3D printing

process in the manuscript. As the function of surface area has been normalized, the relative value of interfacial energies of the three involved interfaces is thus comparable, and the comparison is logical, which can reflect the tendency of the motion of the liquid or solid substance during the printing process.

17. Please state this relation clearly.

9 be completely cured to a predetermined 3D structure only when an appropriate relationship exists
10 between the adhesion properties of the three involved interfaces and the curing interface. The adhesion properties of the S-
11 PDMS are critical as the curing interface for the one-droplet printing process.

Reply: Thanks for the reviewer’s comment. The relation has been discussed and concluded just before this sentence. Accordingly, we revised this sentence to “Therefore, the liquid resin droplet can be cured to a predetermined 3D structure only with an appropriate relationship of $\gamma_1 > \gamma_3$ and $\gamma_3 > \gamma_2$ among the interfacial adhesions for the three involved interfaces” in Line 17, Page 10 in the revised manuscript.

18. Is this the weight of the cured structure to the weight of the droplet until the end of the printing process? If it is nearly 100% could you indicate where the rest of the resin ends up. For example if it is 99.9% where does the 0.1% end up in?

14 before cleaning to the initial liquid resin) is nearly 100% for all tested processes. Considering the
15 wetting behavior and the capillary phenomenon, the weight of the cured structure,
16 liquid resin unavoidable adheres to the supporting plate and the cured structure, and the removal

Reply: Thanks for the reviewer’s comment. The rest of the resin is adhering to the supporting plate. As the interfacial adhesion of liquid resin/cured resin is larger than that of liquid resin/curing interface ($\gamma_1 > \gamma_3$), the TCL of the liquid resin droplet can continuously recede on the curing interface during the continuous printing process. There is thus almost no residual on the curing interface after one-droplet 3D printing, which can be proved by Supplementary Movie 1. As the supporting plate we used is an aluminum plate, some liquid resin will remain on the it through forming liquid bridge between the supporting plate and the cured structure.

Accordingly, we have added the description of “The remaining 0.4% of liquid resin is left on the supporting plate due to the adhesion between liquid resin and supporting plate (as displayed in Figure 1b₃)” in Line 6, Page 4. In addition, the expression of “nearly 100%” has been changed to the specific experimental values of the wet material utilization efficiency.

19. Does this imply that one drop printing allows for the printing of sharper details? That result could be emphasized more. To me that is more important as resin costs.

- 1 Moreover, when the UV projection pattern is varied from a round shape to a V-grooved shape, the dry
- 2 material utilization efficiency decreases greatly for traditional printing, but **it** increases for the
- 3 one-drop printing. Variations in the UV pattern will determine the structure by essentially influencing the contact line morphol
- 4 structure by essentially influencing the contact line morphol

Reply: Thanks for the reviewer's comment. Accordingly, we have compared the detailed morphologies of the asymmetric V-groove structures (with intersection angle of 30°) printed from one-droplet 3D printing (wet material utilization of ~98.6% and net material utilization efficiency of ~92.5%, Figure 3f in the revised manuscript) and from vat polymerization (wet material utilization of ~87.5% and net material utilization efficiency of ~66.7%, Supplementary Figure 13) through Micro-CT characterization.

Figure R9 (Supplementary Figure 18 in the revised Supplementary Information). Micro-CT characterization of the V-grooved structures prepared from vat polymerization and one-droplet 3D printing. **a**, **b** and **c** are inner side

view of the designed, Micro-CT image of V-grooved structure prepared from vat polymerization and Micro-CT image of V-grooved structure prepared from one-droplet 3D printing, respectively. **d**, **e** and **f** are outer side view of the designed, Micro-CT image of V-grooved structure prepared from vat polymerization and Micro-CT image of V-grooved structure prepared from one-droplet 3D printing, respectively. **g**, **h** and **i** are cross-sectional view of the designed, Micro-CT image of V-grooved structure prepared from vat polymerization and Micro-CT image of V-grooved structure prepared from one-droplet 3D printing, respectively. The blue V patterns in **h** and **i** are designed V patterns. Scale bars are 2 mm.

As shown in Figure R9, one-droplet 3D printing process can print sharper inner V-corner than vat polymerization. This can be due to the dragging force (Equation 2 in the manuscript) acting on the liquid resin contact line, which leads to higher net material utilization efficiency and less residual inside the V-corner. However, the residual on the cured structure and cannot be avoided entirely due to the similar compatibility principle, as the compositions and functional groups are similar for monomers and cured monomers. In other words, extra curing due to residual at the inner V-corner can be partly suppressed during one-droplet 3D printing process. Therefore, one-droplet 3D printing process with higher material utilization efficiencies not only can reduce the residual, but also can increase the printing precision.

Accordingly, we have added the experimental data as Supplementary Figure 18 in the revised Supplementary Information with corresponding descriptions: “As shown in Supplementary Figure 18, the one-droplet 3D printing process with higher material utilization efficiencies can print sharper V-grooved structures than vat polymerization, which can be due to the dragging force (Equation 2) acting on the liquid resin receding contact line, where extra curing due to residual can be partly suppressed comparing with vat polymerization” in Line 9, Page 15 the revised manuscript.

20. Could you indicate the level of improvement of accuracy whit numbers?

11 morphology can suppress excessive curing in the x-y plane and maintain the accuracy and stability of
 12 the continuous 3D printing process (Figure S9 in Supplem
 13 Encouraged by the increased material utilization efficiency, the accuracy and stability of a liquid

Reply: Thanks for the reviewer’s comment. According to the reviewer’s suggestion, we directly monitor the *in situ* curing process of vat polymerization and one-droplet printing process in real-time with the same experimental conditions to quantitatively compare the printing precision and evidence the improved precision. Two parameters, including the printing width of the left and right edges which reflects the outer profile of the printed structure, and the ratio (D/D_{design}) of the printed width (D) to the designed width (D_{design}) which reflects the extra curing phenomenon, are characterized for comparison. As displayed in Figure R7a and the red dotted lines in Figure R7c, the printing process is stable with the left and right edges keeping straight during the whole process (left edges range from $-492.9 \mu\text{m}$ to $-507.1 \mu\text{m}$, right edges range from $492.9 \mu\text{m}$ to $500.0 \mu\text{m}$), as well as the stable D/D_{design} ratio (remained between 0.986 and 1.007, the red dotted line in Figure R7d). While for vat polymerization, the printing width becomes nonuniform (left edges range from $-259.1 \mu\text{m}$ to -663.4

μm , right edges range from 445.4 μm to 704.5 μm) with protruding or stepped structures on the sidewall along with the increasing of printing time (Figure R7b and the black dotted lines in Figure R7c). The D/D_{design} ratio also becomes larger (from 0.981 to the maximum of 1.336 during the 500 s continuous curing process) and unstable with the increasing of the time (black dotted line in Figure R7d). The real-time characterization of the two processes is displayed in Supplementary Movie 2.

Figure R7 (Supplementary Figure 6a-b in the revised Supplementary Information and Figure 2h-i in the revised manuscript). Comparison of the continuous printing process of the vat polymerization and the one-droplet 3D printing process (printing speed of 100 $\mu\text{m}/\text{s}$, 500s duration for printing structure of 5 cm high). **a.** Series of optical captures of the one-droplet 3D printing process. **b.** Series of optical captures of the vat polymerization process. **c.** The time-variant width change of the left and right edges on the structure constructed from one-droplet continuous printing, with the vat polymerization process as control. Red and black lines represent the width change curves of the one-droplet printing process and the vat polymerization process, respectively. Circles and squares represent the left and right edges of the printed structure, respectively. **d.** The time-variant D/D_{design} ratio change of the one-droplet printing process, with the vat polymerization process as control. Red and black lines represent D/D_{design} ratio change curves of the one-droplet printing process and the vat polymerization process, respectively. Scale bars are 1 mm.

Accordingly, we have added corresponding experimental results as Figure 2h-i in the manuscript, Supplementary Figure 6 in the revised Supplementary Information and Supplementary Movie 2 to indicate the level of improvement of accuracy whit numbers.

21. Could the argumentation of this section be improved. Also with classical vat polymerization processes these results (appropriate contact and marginal integrity)

7 regions can be visualized, which provides a new strategy to monitor the 3D dewetting process. Thus,
8 one-droplet 3D printing can completely transform a liquid resin droplet into a defined structure with
9 high material utilization efficiency through proper adjustment of the curing interface properties, which
10 can save valuable inks and open a new avenue for on-demand 3D printing.

11 In summary, the concept of interfacial property regul
12 droplet during the UV curing process, which leads t

A screenshot of a reviewer comment box is overlaid on the text. The comment text reads: "Could the argumentation of this section be improved. Also with classical vat polymerization processes these results (appropriate contact and marginal integrity)". The comment box has a yellow border and a close button (X) in the top right corner.

Reply: Thanks for the reviewer’s comment. Based on the reviewer’s above comments, we have conducted new experiments involving the one-droplet printing and the vat polymerization processes to display the strategic advantages of the one-droplet 3D printing, including the unavoidable “afterglow” of current DLP UV light source (Figure R1, Figure R2), the disadvantage of repeated using vat resin of the vat polymerization process (Figure R3, Movie R2 which real-time monitors the repeated utilization of vat resin during the vat polymerization process), the unavailability of vat resin even with a few printing layers high for the vat polymerization process (Figure R4), the characterization of liquid mobility on the curing interface and inside the droplet for the one-droplet printing process (Figure R5 and Figure R6), the quantitative comparison of printing stability and printing precision between vat polymerization and one-droplet 3D printing (Figure R7 - Figure R9), and the strategic versatility (Figure R11) of one-droplet 3D printing to prove the advantage of the one-droplet 3D printing process. Corresponding experimental results have been added in the revised manuscript and revised Supplementary Information. We hope the revised version can improve the argumentation of this section and evidence the strategic advantages of the one-droplet 3D printing strategy.

22. This figure seems to indicate that the interfaces are far apart, as figures c7, e7 and g7 seem to indicate otherwise. If i am correct curing will occur directly above the interface zone (quartz, pdms, air)

Reply: Thanks for the reviewer's comment. In the original Figure 2, the schematic images (7-10) are drafted according to the mechanism of why corresponding surfaces are or not low adhesive. For the F-quartz surface, the surface is smooth, therefore the liquid resin is in full contact with the F-quartz surface, and *in situ* curing process will transform the resin adhesion on the interface from pure liquid-solid adhesion to pure solid-solid adhesion, which is F_{solid} as displayed in **c7**. Here, Figures **c8** to **c10** display the scheme of the *in situ* curing process on the F-quartz surface. For the S-PDMS surface, there is a lubricant layer where the liquid resin is in contact with the composite S-PDMS surface. After *in situ* curing process, the adhesion is the composite of liquid-solid adhesion and solid-solid adhesion as displayed in **e7**. Here, **e8** to **e10** display the scheme of the *in situ* curing process on the S-PDMS surface. For superamphiphobic surface, the low liquid adhesion property is due to the air entrapment among the hierarchical structures, which results in a composite liquid-air-solid interface. After *in situ* curing process, the adhesion is the composite of solid-solid adhesion and solid-gas adhesion, as displayed in **g7**. Here, **g8** to **g10** display the scheme of the *in situ* curing process on the superamphiphobic surface.

Figure R16 (Figure 2c, 2e, 2g in the revised manuscript). The influence of interfacial adhesion properties on the one-droplet 3D printing process.

To make it more easily for readers to understand, we have changed the location of **8** to **10** with **7**, i.e., the original **c₇**, **e₇** and **g₇** are moved to **c₁₀**, **e₁₀** and **g₁₀**, respectively. Correspondingly, the original **c₈** to **c₁₀**, **e₈** to **e₁₀**, and **g₈** to **g₁₀** are put forward, respectively. In addition, the figures about the superamphiphobic surface are changed to **e** and the figures about the S-PDMS surface are changed to **g** in the revised Figure 2, as shown in Figure R16. Furthermore, descriptions of the corresponding figures are revised accordingly.

Captions of supplementary movie for Responses to Reviewers

Movie R1. Real-time monitoring of the curing of the “dark” region which connects to the patterned region.

Movie R2. Real-time monitoring of the repeated utilization of vat resin during the vat polymerization process.

REVIEWERS' COMMENTS:

Reviewer #1 (Remarks to the Author):

I'm happy with the changes and can support publishing the paper in its current form.

Daniel S. Engstrom

Loughborough University, UK

Reviewer #2 (Remarks to the Author):

After revision, the present form of the manuscript is suitable for publication in Nature Communications.